# Store-Operated Ca^2+^ Entry Is Up-Regulated in Tumour-Infiltrating Lymphocytes from Metastatic Colorectal Cancer Patients

**DOI:** 10.3390/cancers14143312

**Published:** 2022-07-07

**Authors:** Pawan Faris, Agnese Rumolo, Laura Tapella, Matteo Tanzi, Alessia Metallo, Filippo Conca, Sharon Negri, Konstantinos Lefkimmiatis, Paolo Pedrazzoli, Dmitry Lim, Daniela Montagna, Francesco Moccia

**Affiliations:** 1Laboratory of General Physiology, Department of Biology and Biotechnology “Lazzaro Spallanzani”, University of Pavia, 27100 Pavia, Italy; faris.pawan@unipv.it (P.F.); alessia.metallo01@universitadipavia.it (A.M.); sharon.negri01@universitadipavia.it (S.N.); 2Department of Biology, College of Science, Salahaddin University, Erbil 44001, Iraq; 3Department of Sciences Clinic-Surgical, Diagnostic and Pediatric, University of Pavia, 27100 Pavia, Italy; agnese.rumolo01@universitadipavia.it; 4Pediatric Clinic, Foundation IRCCS Policlinico San Matteo, 27100 Pavia, Italy; dmitry.lim@uniupo.it; 5Department of Pharmaceutical Sciences, Università del Piemonte Orientale “Amedeo Avogadro”, 27100 Novara, Italy; laura.tapella@uniupo.it; 6UOSD Cell Factory, Pediatric Hematology/Oncology, Foundation IRCCS Policlinico San Matteo, 27100 Pavia, Italy; matteo.tanzi01@universitadipavia.it; 7Department of Molecular Medicine, Human Physiology Unit, University of Pavia, 27100 Pavia, Italy; filippo.conca01@universitadipavia.it (F.C.); konstantinos.lefkimmiatis@unipv.it (K.L.); 8Filippo Conca and Konstantinos Lefkimmiatis, Foundation for Advanced Biomedical Research, Veneto Institute of Molecular Medicine, 35129 Padua, Italy; 9Internal Medicine and Medical Therapy, University of Pavia, 27100 Pavia, Italy; paolo.pedrazzoli@unipv.it; 10Medical Oncology Unit, Fondazione IRCCS Policlinico San Matteo, 27100 Pavia, Italy

**Keywords:** colorectal carcinoma, tumour-infiltrating lymphocytes, store-operated Ca^2+^ entry, diacylglycerol kinase, BTP-2

## Abstract

**Simple Summary:**

Store-operated Ca^2+^ entry (SOCE) has long been known to regulate the differentiation and effector functions of T cells as well as to be instrumental to the ability of cytotoxic T lymphocytes to target cancer cells. Currently, no information is available regarding the expression and function of SOCE in tumour-infiltrating lymphocytes (TILs) that have been expanded in vitro for adoptive cell therapy (ACT). This study provides the first evidence that SOCE is up-regulated in ex vivo-expanded TILs from metastatic colorectal cancer (mCRC) patients. The up-regulation of SOCE mainly depends on diacylglycerol kinase (DGK), which prevents the protein kinase C-dependent inhibition of Ca^2+^ entry in normal T cells. Of note, the pharmacological blockade of SOCE with the selective inhibitor, BTP-2, during target cell killing significantly increases cytotoxic activity at low TIL density, i.e., when TILs-mediated cancer cell death is rarer. This study, albeit preliminary, could lay the foundation to propose an alternative strategy to effect ACT. It has been shown that ex vivo-expanded TILs did not improve the disease-free survival rate in mCRC patients. Our results strongly suggest that pre-treating autologous TILs with a SOCE or DGK inhibitor before being infused into the patient could improve their cytotoxic activity against cancer cells.

**Abstract:**

(1) Background: Store-operated Ca^2+^ entry (SOCE) drives the cytotoxic activity of cytotoxic T lymphocytes (CTLs) against cancer cells. However, SOCE can be enhanced in cancer cells due to an increase in the expression and/or function of its underlying molecular components, i.e., STIM1 and Orai1. Herein, we evaluated the SOCE expression and function in tumour-infiltrating lymphocytes (TILs) from metastatic colorectal cancer (mCRC) patients. (2) Methods: Functional studies were conducted in TILs expanded ex vivo from CRC liver metastases. Peripheral blood T cells from healthy donors (hPBTs) and mCRC patients (cPBTs) were used as controls. (3) Results: SOCE amplitude is enhanced in TILs compared to hPBTs and cPBTs, but the STIM1 protein is only up-regulated in TILs. Pharmacological manipulation showed that the increase in SOCE mainly depends on tonic modulation by diacylglycerol kinase, which prevents the protein kinase C-dependent inhibition of SOCE activity. The larger SOCE caused a stronger Ca^2+^ response to T-cell receptor stimulation by autologous mCRC cells. Reducing Ca^2+^ influx with BTP-2 during target cell killing significantly increases cytotoxic activity at low target:effector ratios. (4) Conclusions: SOCE is enhanced in ex vivo-expanded TILs deriving from mCRC patients but decreasing Ca^2+^ influx with BTP-2 increases cytotoxic activity at a low TIL density.

## 1. Introduction

Colorectal carcinoma (CRC) represents the third most common cause of malignancy-associated mortality, which is mainly due to the development of liver metastases [1]. Surgical resection represents the only potentially curative treatment in combination with adjuvant or perioperative chemotherapy [1]. Unfortunately, metastatic recurrence affects ≈80% of the patients [1], the majority of whom succumb to disease progression [2]. Therefore, the design of a more effective strategy to fully eradicate liver metastases is a compelling need for metastatic CRC (mCRC) patients.

Adoptive cell therapy (ACT) represents a promising approach to achieve complete and durable cancer eradication by the transfer of ex vivo-expanded cells reactive against tumour [3,4]. Studies on metastatic melanoma demonstrated that autologous tumour-infiltrating lymphocytes (TILs), ex vivo-expanded in the presence of interleukin-2 (IL-2) and then reinoculated into the patients, may be exploited to induce a long-lasting (up to 3 years) disease regression [5,6]. The ability of TILs to induce a complete response and promote tumour regression was recently reported also for cervical carcinoma [7]. CD8+ cytotoxic T lymphocytes (CTLs), which are induced to recognise specific tumour-associated antigens (TAAs), have also been shown to be effective against types of solid tumours [8,9,10,11,12]. The short-term expansion of naturally occurring tumour-reactive CD8+ TILs from the liver metastases of CRC patients has recently been described by two independent groups [13,14]. However, it is still unclear whether these TILs could be employed to effectively target metastatic CRC (mCRC) [15]. Unravelling whether the intracellular signalling pathways that are recruited downstream of T-cell receptors (TCR) are deranged is mandatory to fully exploit the therapeutic potential of expanded TILs against cancer cells.

An increase in intracellular Ca^2+^ concentration ([Ca^2+^]_i_) is of paramount importance to promote the differentiation and effector functions of T cells by driving processes that are key to the immune response, including proliferation, differentiation, cytokine secretion and cytotoxicity [16,17]. TCR engagement results in the activation of phospholipase C-1γ1 (PLCγ1) that cleaves phosphatidylinositol-4,5-bisphosphate (PIP_2_) into inositol-1,4,5-trisphosphate (InsP_3_) and diacylglycerol (DAG). InsP_3_ binds to and gates the ionotropic InsP_3_ receptors (InsP_3_Rs), which mediate Ca^2+^ release from the endoplasmic reticulum (ER), the largest endogenous Ca^2+^ reservoir in T cells. The following reduction in ER Ca^2+^ concentration ([Ca^2+^]_ER_) is detected by Stromal Interaction Molecule 1 (STIM1), which serves as sensor of [Ca^2+^]_ER_ and is thereafter prompted to translocate from perinuclear to peripheral ER cisternae, where STIM1 traps and activates the Ca^2+^-selective Orai1 channels at the plasma membrane [16,18,19]. The following influx of Ca^2+^, known as store-operated Ca^2+^ entry (SOCE), induces the nuclear translocation of the Ca^2+^-dependent nuclear factor of activated T cells c2 (NFATc2 or NFAT1), cytokine production and cytotoxicity [16,17,20]. Therefore, ex vivo-expanded TILs are predicted to impinge on a functional SOCE to be therapeutically effective against cancer cells. In accord, the pharmacological or genetic deletion of SOCE impairs the cytotoxic activity of human [21,22] and murine [23] CTLs, thereby favouring the engraftment of cancer cells and promoting tumour growth [23]. Surprisingly, although neoplastic transformation may induce a dramatic derangement in the expression and/or function of the SOCE machinery [24,25,26,27,28,29,30], it is still unknown whether a functional SOCE is expressed in TILs that are generated for ACT. Herein, we addressed this compelling issue by focusing on ex vivo-expanded TILs isolated from the liver metastases of mCRC patients. Peripheral blood T (PBT) cells from healthy donors (hPBTs) and mCRC patients (cPBTs) were used as controls, as recently suggested [31,32].

## 2. Materials and Methods

### 2.1. Patients and Donors

This study was approved by the Institutional Review Board of the Foundation IRCCS Policlinico San Matteo (Protocol Number 20190069408). After signing an informed consent form, 10 patients (>18 years) affected by mCRC who had undergone surgery to remove liver metastases were enrolled (Appendix A). All procedures were performed according to the guidelines prescribed for the treatment of CRC neoplasia, and no patient was subjected to unnecessary invasive procedures. Peripheral blood cells were derived from the same mCRC patients (cPBT) at the time of surgery or from the buffy coats of 10 healthy donors (hPBT). Buffy-coat units were released by the Blood Bank of SIMT, Policlinico San Matteo, Pavia, according to Article 8 of the Legislative Decree 2, November 2015, of the Italian Department of Health, with blood donor-specific informed consent implemented by the Blood Bank Conference of Regione Lombardia.

### 2.2. Isolation of TILs and Tumour Cells

Tumour specimens (liver metastases) from 10 patients (Appendix A) were processed as previously described [9] with the GentleMACS Dissociator (Miltenyi Biotec, Bologna, Italy, cat# 130-093-235) after being treated with the Tumor Dissociation Kit (Miltenyi Biotech, cat# 130-095-929). After this procedure, the tumour cells were filtered (Cell Strainers, Falcon, Corning Stone, Staffordshire, England, cat# 352350) to remove clusters and to obtain a single-cell suspension before magnetic separation. The cells were then checked for viability by trypan blue dye exclusion and resuspended in buffer containing phosphate-buffered saline (PBS, Euroclone, Pero, Italy, cat# ECB4004L) pH 7.2, 0.5% bovine serum albumin (BSA, Kedrion Biopharma, Barga, Italy) for the isolation of CD45-positive cells, using human CD45 microbeads (cat# 130-118-780) and an LS Column (cat# 130-042-401) (Miltenyi Biotec), according to the manufacturers’ instructions. Negative and positive cells were collected, centrifuged and checked for viability and recovery.

### 2.3. Expansion of mCRC Cells

Negative cells, which contain tumour cells, were plated in 12-well plates at a concentration of 0.5–1 × 10^6^ cells/mL of CellGro SCGM (Cell Genix, Freiburg, Germany, cat# 20802-0500), supplemented with 20% foetal bovine serum (FBS) (Euroclone, cat# ECS0180D) and 0.1% gentamicin (Gibco, Life Technologies Limited, Paisley, UK, cat# 15750-037), and cultured in 25 cm^2^ tissue flasks (Corning Stone, Staffordshire, UK) at 37 °C and 5% CO_2_. Viable tumour cells attached to the flask within 12–24 h. At the first medium change, the cells were put into a fresh flask. The cells at 75% to 100% confluence were detached with 0.25% trypsin and 0.02% EDTA (Euroclone) in a calcium/magnesium-free balanced solution. The culture medium was changed twice a week and cellular homogeneity was evaluated microscopically every 24–48 h. Tumour cells derived from early passages (3–5) cultures were cryopreserved in 90% FBS and 10% DMSO (Alchimia, Ponte San Nicolò, Italy, cat# CRN 001-00) and stored in liquid nitrogen for further experiments. To confirm the neoplastic origin of the cultured cells, the cells underwent morphological and immunocytochemical analysis, as previously described [9,26]. We were able to isolate and grow tumour cells from all 10 patients enrolled in this study.

### 2.4. TIL Expansion and Characterisation

We designed a methodological approach to optimise TIL expansion from mCRC samples after the isolation of lymphoid cells using CD45 TIL Microbeads (Miltenyi Biotec). CD45+ cells were rapidly expanded for 14 days at 37 °C in six-well plates (Corning Stone) in the CellGRO medium in the presence of OKT3 (30 ng/mL) (Miltenyi Biotech, cat# 170-076-124), IL-15 (10 ng/mL) (Miltenyi Biotech, cat# 170-076-114), IL-2 (3000 U) (Novartis, Basilea, Switzerland, cat# AIC 02713110/M) and irradiated (50 Gy) feeder cells. On days 7 and 10, the cells were monitored, counted and split with medium containing IL-2 and IL-15; on day 14, the cells were checked for viability and then cryopreserved for further characterisation. Based upon our preliminary experiments aimed to optimise the expansion method, we documented that a 14-day culture represents the best timepoint for the expansion rate and vitality of recovered TILs; further expansion might reduce the number of viable cells.

The TILs were analysed by immunostaining with specific monoclonal antibodies, followed by analysis using flow cytometry (BD FACS Canto II, Beston, Dickinson and Company, BD Biosciences, Franklin Lakes, NJ, USA), as previously described [9]. The following conjugated monoclonal antibodies were used: anti-CD45, anti-CD3 (cat# IM2467), anti-CD4 (cat# 651849), anti-CD8 (cat# 641400) and anti-CD56 (cat# 345810) (Beckman Coulter, Brea, CA, USA & BD Biosciences). The cytotoxic capacity of the TILs was evaluated in a 5 h cytotoxicity assay against the colorectal carcinoma SW480 cell line (obtained from ATCC, Sesto San Giovanni, Milan, Italy) and against autologous mCRC cells labelled with Chromium-51 (^51^Cr). Briefly, SW 480 cells were labelled overnight (ON) in 200 µL with 20 µL of ^51^Cr, whereas adherent, non-confluent mCRC cells were washed and then labelled ON with 20 µL of ^51^Cr. The day after, the SW 480 cells were washed, while the mCRC cells were detached and washed. The SW480 and mCRC cells were then added to TILs at varying effector:target (E:T) ratios. After 5 h incubation at 37 °C, supernatants were harvested for the quantification of chromium release. As a control, we employed donor-derived anti-leukaemia CTL generated and expanded using a methodological approach that includes leukaemia-specific stimulations followed by a rapid expansion using the same protocol employed for TILs [33].

### 2.5. Total RNA Extraction and Real-Time PCR

Total mRNA was extracted from 0.5 × 10^6^ cells using a SPLIT RNA Extraction Kit (Lexogen, Austria, cat# 008.48) according to the manufacturer’s instructions, as described in [34]. cDNA was synthesised from 0.5–1 µg of total RNA using a SensiFAST cDNA Synthesis Kit (BioLine, London, UK, Cat. BIO-65054). Real-Time PCR was performed using iTaq qPCR master mix according to the manufacturer’s instructions (Bio-Rad Laboratories, Hercules, CA, USA; cat# 1725124) on an SFX96 Real-Time system (Bio-Rad Laboratories). An S18 ribosomal subunit was used to normalise the raw data. The data are calculated for each sample as 2^-(Ct(GOI) − Ct(S18)) and expressed as mean ± standard deviation from three independent cultures, each run in triplicate. The primers’ sequences are provided in Appendix A.

### 2.6. Immunoblotting

The cells were lysed with 100 µL of lysis buffer (50 mM Tris-HCl, pH = 7.4), sodium dodecyl sulphate (SDS) 0.5%, 5 mM EDTA, 10 µL of protease inhibitors cocktail (PIC) (Millipore, cat# 539133) and phosphatase inhibitor cocktail (ThermoFisher Scientific, Waltham, MA, USA; cat# 78428) and collected in a 1.5 mL tube. The lysates were boiled at 96 °C for 5 min and then quantified with a QuantiPro BCA Assay Kit (Sigma-Aldrich, Milan, Italy; cat# SLBF3463). Then, 10 µg (for STIM1) and 20 µg (for ORAI1) of proteins were mixed with the right amount of Laemmli Sample Buffer 4X (Bio-Rad Laboratories), and boiled. The samples were then loaded onto a 12% polyacrylamide–sodium dodecyl sulphate gel for SDS-PAGE. The proteins were transferred onto nitrocellulose membrane, using Mini Transfer Packs with Trans-Blot^®^ Turbo TM (Bio-Rad Laboratories) according to manufacturer’s instructions (Bio-Rad Laboratories). The membranes were blocked in 5% skimmed milk (Sigma, cat# 70166) for 45 min at room temperature. Subsequently, the membranes were incubated with indicated primary antibody, overnight, at 4 °C. The primary antibodies used are listed in Appendix A; anti-β-Actin was used to normalise the protein loading.

### 2.7. Solutions

The physiological salt solution (PSS) had the following composition: 150 mM NaCl, 6 mM KCl, 1.5 mM CaCl_2_, 1 mM MgCl_2_, 10 mM glucose, 10 mM HEPES. In Ca^2+^-free solution (0Ca^2+^), Ca^2+^ was replaced with 2 mM NaCl and 0.5 mM EGTA was added. The solutions were titrated to pH 7.4 with NaOH. The osmolality of the extracellular solution, as measured with an osmometer (Wescor 5500), was 300–310 mmol/kg.

### 2.8. [Ca^2+^]_i_ Measurements

The T cells were loaded with 2 µM Fura-2 acetoxymethyl ester (Fura-2/AM; 1 mM stock in dimethyl sulfoxide) in PSS for 30 min at 37 °C, as previously shown for mCRC cells [26]. After washing in PSS, the coverslip was fixed to the bottom of a Petri dish and the cells observed by an upright epifluorescence Axiolab microscope (Carl Zeiss, Oberkochen, Germany), usually equipped with a Zeiss ×40 Achroplan objective (water-immersion, 2.0 mm working distance, 0.9 numerical aperture). The T cells were excited alternately at 340 and 380 nm, and the emitted light was detected at 510 nm. A first neutral density filter (1 or 0.3 optical density) reduced the overall intensity of the excitation light, and a second neutral density filter (optical density = 0.3) was coupled to the 380 nm filter to approach the intensity of the 340 nm light. A round diaphragm was used to increase the contrast. The excitation filters were mounted on a filter wheel (Lambda 10, Sutter Instrument, Novato, CA, USA). Custom software, working in the LINUX environment, was used to drive the camera (Extended-ISIS Camera, Photonic Science, Millham, UK) and the filter wheel, and to measure and plot online the fluorescence from 10 up to 40 rectangular “regions of interest” (ROI). Each ROI was identified by a number. Since the cell borders were not clearly identifiable, an ROI may not include the whole cell or may include part of an adjacent cell. Adjacent ROIs never superimposed. [Ca^2+^]_i_ was monitored by measuring, for each ROI, the ratio of the mean fluorescence emitted at 510 nm when exciting alternately at 340 and 380 nm (shortly termed “ratio”). An increase in [Ca^2+^]_i_ causes an increase in the ratio [26]. Ratio measurements were performed and plotted online every 3 s. The experiments were performed at room temperature (22 °C). Resting [Ca^2+^]_i_ in the three cell types was monitored by using the Grynkiewicz method, as described in [35].

### 2.9. Statistics

All of the Ca^2+^ imaging data were collected from T cells deriving from three donors. Tumour-derived T cells, i.e., cPBTs and TILs, isolated from all 10 patients were used throughout the study. Therefore, cPBTs and TILs expanded from three distinct donors could be used in different experimental conditions. Likewise, the hPBTs isolated from three healthy donors were used for Ca^2+^ imaging experiments. Pooled data are given as mean ± SE, while the number of cells analysed is indicated between parentheses. Statistical significance (*p* < 0.05) was evaluated using the Student’s *t*-test for unpaired observations and the one-way ANOVA test followed by the post hoc Bonferroni test, as appropriate.

### 2.10. Chemicals

Fura-2/AM was obtained from Invitrogen (Life Technologies; cat# F1221). BTP-2 (cat# 203890) and myo-inositol 1,4,5-trisphosphate hexakis (butyryloxymethyl) (InsP_3_BM) (cat# 3-1-145) were obtained from Calbiochem (Merk Millipore). All other chemicals were of analytical grade and were obtained from Sigma-Aldrich: cyclopiazonic acid (CPA) (cat# C1530), ATP (cat# A2383), caffeine (cat# C0750), R59949 (cat# D5794), ritanserin (cat# R103), phorbol 12-myristate 13-acetate (PMA) (cat# 79346), Gö-6983 (cat# G1918), U73122 (Cat# U6756, 2-Aminoethyl diphenylborinate (2-APB) (cat# D9754) and ionomycin (cat# I0634).

## 3. Results

### 3.1. Characterisation of Ex Vivo-Expanded TILs

The cultures of the TILs obtained from the 10 patients enrolled in this study were evaluated after 14 days for surface antigens and potency, in order to confirm that the cells expanded from the tumour samples were T cells specifically directed against cancer cells. The great majority of the TILs were CD3+ cells (>96%), containing both CD3+/CD4+ and CD3+/CD8+ cells (mean 60%, SD 18% and mean 34%, SD 22%, respectively) (Figure 1A). Negligible percentages of CD3−/CD56+ (less than 5%) cells were documented (Figure 1A). The dot blots of two representative patients are shown in Appendix A. The potency of the TILs was evaluated against both the SW480 cells lines and autologous mCRC cells obtained from all 10 patients (Figure 1D). The TIL potency was evaluated in a 5-h cytotoxicity assay based on ^51^Cr release by the target cells. All TILs displayed high levels of cytotoxicity against SW480 cells (mean: 56% and SD 7% at effector:target (E:T) ratio of 50:1), which were maintained even at lower E:T ratios. The TILs were also able to efficiently kill autologous tumour cells as well (mean: 41% and SD 8% at E:T ratio of 50:1), despite being slightly lower compared to the immortalised cell line, which is also maintained even at lower E:T ratios. These data documented that the cytotoxic activity was not due to CD3−/CD56+ NK cells, which were almost completely absent from the expanded TILs, as shown in Figure 1A, confirming that the TILs were specifically directed against mCRC cells, both in the autologous and immortalised cell line (Figure 1B).

### 3.2. SOCE Is Larger in Ex Vivo-Expanded TILs

The remodelling of the Ca^2+^ handling machinery in TILs, compared to hPBTs and cPBTs, was evaluated in cells loaded with the Ca^2+^-sensitive fluorophore Fura-2/AM. The resting [Ca^2+^]_i_ was significantly (*p* < 0.001) higher in the TILs than in the cPBTs and hPBTs (Appendix A). The “Ca^2+^ add-back” protocol was then exploited to simultaneously evaluate ER Ca^2+^ release and SOCE in the three cell types. As described elsewhere [23,26], the cells were challenged with CPA (30 µM), a selective inhibitor of Sarco-Endoplasmic Reticulum Ca^2+^-ATPase (SERCA) activity, in the absence of extracellular Ca^2+^ (0Ca^2+^) to induce ER Ca^2+^ release and thereby promote ER Ca^2+^ depletion. Extracellular Ca^2+^ was restored upon recovery to the baseline to selectively measure the SOCE. Careful analysis of the Ca^2+^ tracings (Figure 2A) revealed that the amplitude of CPA-induced ER Ca^2+^ release was as follows: cPBTs > TILs > hPBTs (Figure 2B). However, the CPA-evoked SOCE was significantly (*p* < 0.001) larger in the TILs than in cPBTs, which, in turn, exhibited a larger SOCE than hPBTs (Figure 2B). In keeping with these observations, when the T cells were challenged with CPA (30 µM) in the presence of extracellular Ca^2+^ (1.5 mM), the amplitude of the Ca^2+^ peak was as follows: TILs >> cPBTs > hPBTs (Appendix A). Pre-incubation with BTP-2 (20 µM), which selectively inhibits Orai1 in hPBTs [36] and CTLs [23], significantly (*p* < 0.001) reduced CPA-evoked SOCE in the TILs (Figure 2C,D). Therefore, Orai1 channels also mediate SOCE in these cells. These pieces of evidence indicate that ex vivo-expanded TILs undergo a remodelling of the Ca^2+^-handling machinery that leads to SOCE hyperactivation.

### 3.3. InsP_3_Rs- and RyRs-Mediated Ca^2+^ Release Result in Enhanced SOCE in Ex Vivo-Expanded TILs

Ca^2+^ release from the ER is mediated by InsP_3_Rs and ryanodine receptors (RyRs), which can both result in SOCE activation in hPBTs [16,17]. The InsP_3_-induced intracellular Ca^2+^ mobilisation was first evaluated by stimulating the cells with the InsP_3_-synthesising autacoid, ATP (100 µM), as shown in other types of cancer cells [26,38]. Preliminary experiments confirmed that the ATP-induced intracellular Ca^2+^ mobilisation was mediated by InsP_3_Rs in hPBTs (Appendix A). The ATP-induced ER Ca^2+^ release did not differ among the three different cell types (Figure 3A,B). However, SOCE amplitude in the TILs and cPBTs was similar and significantly (*p* < 0.001) higher than hPBTs (Figure 3A,B). When the T cells were challenged with ATP (100 µM) in the presence of extracellular Ca^2+^ (1.5 mM), the amplitude of the Ca^2+^ peak was as follows: TILs = cPBTs > hPBTs (Appendix A).

To circumvent the G_q/11_ protein–PLCβ-InsP_3_-signalling pathway recruited downstream of purinergic P_2Y_ receptors, InsP_3_Rs were directly activated with a membrane-permeant esterified form of InsP_3_, known as InsP_3_BM [38]. The two components (ER Ca^2+^ release and SOCE) of the Ca^2+^ response to InsP_3_BM (1 µM) displayed a similar trend as the one described for ATP: no statistically relevant difference in ER Ca^2+^ release among the three cell types, while SOCE amplitude was as follows: TILs > cPBTs > hPBTs (Figure 3C,D).

The RyRs-mediated Ca^2+^ signals were compared by stimulating the cells with caffeine (5 mM), a selective RyR agonist [39]. The caffeine-evoked ER Ca^2+^ release did not differ between the hPBTs and TILs, but it was significantly (*p* < 0.001) larger in cPBTs (Figure 3E,F). However, the caffeine-evoked SOCE in the cPBTs and TILs was still significantly (*p* < 0.001) larger than in the hPBTs (Figure 3E,F).

Overall, these findings demonstrate that SOCE amplitude is always enhanced in ex vivo-expanded TILs, whatever the stimulus inducing ER Ca^2+^ depletion (CPA, ATP, InsP_3_BM and caffeine), although the distinct ER Ca^2+^ sub-compartments (InsP_3_Rs vs. RyRs) underwent different patterns of remodelling in TILs vs. cPBTs (i.e., selective increase in RyRs-mediated ER Ca^2+^ release in cPBTs).

### 3.4. Molecular Characterisation of the Ca^2+^ Toolkit Reveals the Up-Regulation of STIM1 Protein in Ex Vivo-Expanded TILs

In order to gain further insights into the molecular mechanisms underlying the remodelling of the Ca^2+^-handling machinery in the TILs, we first performed a thorough qRT-PCR analysis of the mRNAs harvested from the three different cell types. Figure 4A shows that InsP_3_R1, RyR1, STIM1 and Orai1 were the predominant isoforms expressed in hPBTs, in agreement with previous observations [18,19,41,42,43,44]. Subsequent transcriptomic analysis, which is presented in Appendix A, revealed that InsP_3_R1, but not InsP_3_R2 and InsP_3_R3, was significantly (*p* < 0.001) downregulated in TILs compared to hPBTs and cPBTs. RyR1 was less expressed in cPBTs and TILs compared to hPBTs, while RyR2 was similarly expressed in the three cell types (Appendix A). Likewise, there were no significant changes in the expression levels of the transcripts encoding for SERCA2B (Appendix A), the major SERCA isoforms in T cells [17]. The comparison of ΔCt values of the mRNAs obtained through qRT-PCR did not reveal any significant difference in the expression levels of Orai1 and Orai3, as shown in Appendix A. Conversely, the levels of STIM1 mRNA were as follows: hPBTs > cPBTs > TILs (Appendix A). These transcriptomic data were somehow contrasting with the evidence reported above that the SOCE amplitude increases during the progression from hPBTs to cPBTs and then TILs. Therefore, we carried out a Western blot analysis of the relative changes in the expression of STIM1 and Orai1 proteins, which affect SOCE in hPBTs [18,19,41,43]. The immunoblots revealed a major band of ≈90 kDa for STIM1 (Figure 4B) and of ≈33 kDa for Orai1 (Figure 4B), as expected by their molecular weight [26,43]. The STIM1 protein was modestly, but significantly, up-regulated in TILs compared to hPBTs and cPBTs (Figure 4C). Conversely, there was no difference in the expression levels of the Orai1 protein (Figure 4C). These findings suggest that STIM1 up-regulation could enhance SOCE in TILs, but not in cPBTs, in which the protein expression of the ER Ca^2+^ sensor is not increased. Thus, an additional mechanism could regulate SOCE amplitude in cPBTs and, possibly, in ex vivo-expanded TILs.

### 3.5. Enhanced SOCE in TILs and cPBTs Is Maintained by Diacylglycerol Kinase (DGK) Activity

DGK limits diacylglycerol (DAG) signalling in hPBTs by converting DAG into phosphatidic acid. Interestingly, DGK function and/or expression are up-regulated in TILs, thereby attenuating T-cell cytotoxicity against cancer cells [45,46]. Moreover, SOCE can be inhibited by protein kinase C (PKC) [47,48], which is a downstream target of DAG [45]. Therefore, the pharmacological blockade of DGK activity is predicted to reinstate PKC-mediated inhibition and reduce the SOCE amplitude in neoplastic T cells. Figure 5A shows that CPA-evoked SOCE is not altered by two established DGK inhibitors, i.e., ritanserin (40 µM) and DGKiII (5 µM; also known as R59949) [45,46], in hPBTs. However, SOCE was significantly (*p* < 0.001) increased by inhibiting the basal PKC activity with Gö-6983 (10 µM) (Figure 5A), whereas it was significantly (*p* < 0.001) reduced by directly activating PKC with PMA (10 µM) (Figure 5A). The statistical analysis of these data is reported in Figure 5B. Overall, these preliminary findings revealed that PKC is able to inhibit SOCE in hPBTs, but that its activity is unlikely to be limited by basal DGK activity. Conversely, ritanserin (40 µM) and R59949 (5 µM) dramatically reduced CPA-evoked SOCE in both cPBTs (Figure 5C) and TILs (Figure 5E). Interestingly, Gö-6983 (10 µM) did not remarkably affect SOCE either in cPBTs (Figure 5C) or in TILs (Figure 5E), whereas PMA (10 µM) exerted a significant (*p <* 0.001) inhibitory effect in cPBTs (Figure 5C) and in TILs (Figure 5E). The statistical analysis of these data is reported in Figure 5D,F. Overall, these data strongly suggest that SOCE amplitude in TILs and cPBTs is larger than in hPBTs due to the DGK-positive input that dampens the basal inhibition operated by PKC.

### 3.6. TCR Stimulation Induced a Larger Increase in [Ca^2+^]_i_ in TILs

In order to assess the impact of the increase in SOCE on TCR stimulation, the different types of T cells were challenged with soluble antibodies direct against CD3, which transduces antigen recognition by the variable component of TCR into an increase in [Ca^2+^]_i_ [21,41,49]. One µg/mL anti-CD3 monoclonal antibody (mAB) triggered a stronger oscillatory Ca^2+^ response in TILs than in cPBTs and hPBTs (Figure 6A–C). Although there was no significant difference in the percentage of oscillating cells (Figure 6D), the area under the curve (AUC) corresponding to the intracellular Ca^2+^ signals evoked by TCR stimulation (Figure 6E) and the oscillation frequency (Figure 6F) were significantly larger in the TILs than in the hPBTs and cPBTs. The pharmacological blockade of Orai1 with BTP-2 (20 µM) dampened the spiking Ca^2+^ response (Figure 6G) and significantly (*p* < 0.001) hindered the AUC of the Ca^2+^ signal (Figure 6H) evoked in TILs by 1 µg/mL anti-CD3 antibody.

The “Ca^2+^ add-back” was then exploited to further assess the role of SOCE in the stronger Ca^2+^ response to anti-CD3 stimulation in ex vivo-expanded TILs. Statistical analysis of the Ca^2+^ tracings (Figure 7A) revealed that the anti-CD3 antibody evoked a similar endogenous Ca^2+^ response in the TILs to the hPBTs, while the intracellular Ca^2+^ mobilisation was surprisingly smaller in the cPBTs (Figure 7B). Conversely, SOCE was remarkably larger in the TILs than in the hPBTs and cPBTs (Figure 7B). Overall, these findings demonstrate that a larger SOCE results in a stronger oscillatory Ca^2+^ response to anti-CD3 stimulation in ex vivo-expanded TILs.

### 3.7. SOCE Tunes the Amplitude of the Ca^2+^ Response during Target Cell Killing

TCR is physiologically stimulated by the formation of the immune synapse between cPBTs or TILs with the target cancer cell to induce an increase in [Ca^2+^]_i_ that supports their cytotoxic activity [21,31,50,51]. T cells were adhered to glass substrates, loaded with Fura-2/AM and mCRC cancer cells were then dropped onto them. The preliminary experiments conducted with the SW480 mCRC cell line revealed that 1 × 10^4^ cells/mL was the most suitable cell density to evoke robust intracellular Ca^2+^ oscillations in the TILs (Appendix A). When the adherent cells were challenged with autologous (i.e., deriving from the same patient) mCRC cells, we observed three main intracellular Ca^2+^ signatures: (1) 46.08% (106/230) of responding cells presented intracellular Ca^2+^ oscillations that persist for about 40 min from the beginning of the stimulation (termed “oscillatory pattern”) (Figure 8A); (2) 34.7% (80/230) of the responding cells displayed two slow, intracellular Ca^2+^ waves that again lasted for about 40 min (termed “slow pattern”) (Figure 8B); and (3) 19.1% (44/230) of responding cells exhibited a fast Ca^2+^ transient followed by a slow increase in [Ca^2+^]_i_ that ultimately achieved a plateau phase (therefore termed “plateau pattern”) (Figure 8C), which declined to resting Ca^2+^ levels upon washout of the mCRC cells. When the same experiment was repeated in the same day using the same batches of TILs and mCRC cells, we observed that the pharmacological blockade of SOCE with BTP-2 (20 µM, 10 min) induced all of the TILs to display the “oscillatory pattern” (141/141 responding cells; 100%) (Figure 8D), thereby abolishing the “slow” and “plateau” Ca^2+^ signatures. The AUC corresponding to the intracellular Ca^2+^ signals evoked by TCR stimulation confirmed that BTP-2 inhibited the amount of Ca^2+^ liberated during the immune response (Figure 8E), whereas it increased oscillation frequency (Figure 8F), which reflects the disappearance of the “slow” and “plateau” Ca^2+^ waveforms. In a parallel set of experiments, the cytotoxic activity was measured in the absence (Ctrl) and presence of BTP-2 (20 µM) using the same batches of cells that were employed for the Ca^2+^ imaging recordings. Surprisingly, the average cytotoxic activity against autologous mCRC cells at each E:T ratio was increased by pre-treating the ex vivo-expanded TILs with BTP-2 (20 µM) (Figure 8G), with the increase becoming statistically significant (*p* < 0.05) at low E:T ratios (Figure 8H). Overall, these findings demonstrate that reducing Ca^2+^ influx with BTP-2 during target cell killing significantly increases the cytotoxic activity at a low TIL density, i.e., when TILs-mediated cancer cell death is rarer.

## 4. Discussion

SOCE is a crucial signalling pathway to stimulate T cells against cancer cells [22,23,52]. Therefore, unravelling whether the SOCE machinery is deranged and/or whether it is uncoupled from the downstream Ca^2+^-dependent cytotoxic response against cancer cells is of paramount importance to help improve the therapeutic outcome of ACT via autologous TILs. Herein, we demonstrated, for the first time, that SOCE is up-regulated in TILs and cPBTs deriving from mCRC patients compared to hPBTs, mainly due to tonic DGK-dependent modulation. Surprisingly, the increase in SOCE amplitude is associated with a reduction in TILs’ cytotoxic activity against autologous mCRC cells compared to that of CTLs. This unexpected finding could explain their limited therapeutic efficacy in the patients [53].

Neoplastic transformation is associated with a dramatic remodelling of the Ca^2+^-handling machinery [24,25], which does not only involve tumour cells, but also other cellular constituents of the tumour microenvironment [54]. Notably, SOCE is strongly downregulated in fresh TILs from a number of tumours due to the inhibitory effect of programmed death-1 (PD-1) [31,32], but SOCE amplitude quickly recovers after tumour dissociation [31]. Nevertheless, no information is currently available regarding the molecular remodelling and the role of SOCE in TILs that have been expanded in culture to be reinoculated within the patient. This gap in the knowledge is quite surprising, as intracellular Ca^2+^ dynamics, particularly SOCE activation, are crucial for CTLs to mount the immune response against cancer cells upon T antigen presentation by the major histocompatibility complex [22,23]. In accord, STIM1- and Orai1-deficient subjects displayed a significant propensity to develop human herpesvirus-8-dependent Kaposi sarcoma [55] and Epstein–Barr virus-associated B-cell lymphoma [56]. Herein, we adopted a protocol that involves restoring Ca^2+^ to the extracellular solution after depletion of the ER Ca^2+^ store with the specific SERCA inhibitor, CPA, to separately evaluate ER Ca^2+^ release and SOCE activation [26,38]. By doing so, we found that SOCE was larger in TILs than in hPBTs and cPBTs, which is the most likely explanation for their larger resting [Ca^2+^]_i_ [17]. Enhanced SOCE could reflect larger ER Ca^2+^ depletion and/or enhanced STIM/Orai protein expression. However, the ER Ca^2+^-releasing ability of TILs, as monitored by CPA-evoked intracellular Ca^2+^ mobilisation, was not increased compared to cPBTs and hPBTs. In T cells, SOCE can be elicited by ER Ca^2+^ depletion following intraluminal Ca^2+^ efflux through either InsP_3_Rs or RyRs [16,17]. However, ER Ca^2+^ mobilisation via InsP_3_Rs (physiologically activated with ATP or directly engaged with InsP_3_-BM) and RyRs (pharmacologically stimulated with caffeine) was never enhanced in TILs, although SOCE was always more up-regulated compared to hPBTs. The lack of a clear relationship between the extent of the ER Ca^2+^ mobilisation and SOCE activation reported in the present investigation is consistent with the long-standing notion that SOCE is not tightly coupled to store emptying [57] and that discrete ER subdomains can effectively stimulate SOCE upon direct stimulation of InsP_3_Rs and RyRs in hPBTs [41]. Overall, these preliminary findings demonstrated that in in vitro-expanded TILs: (1) SOCE is operative, (2) can be engaged by InsP_3_Rs- and RyRs-mediated depletion of the ER Ca^2+^ store, and (3) is up-regulated compared to hPBTs. Therefore, it would be tempting to conclude that the molecular mechanism responsible for SOCE up-regulation in TILs is the same as in cPBTs, a fraction of which is recruited to liver metastases and is likely to represent the main cellular population expanded in vitro. This hypothesis was supported by the notion that the SOCE, activated upon the physiological depletion of the ER Ca^2+^ pool via InsP_3_Rs and RyRs, is also up-regulated in cPBTs compared to hPBTs.

Surprisingly, the molecular analysis of STIM and Orai expression by qRT-PCR and immunoblotting revealed that STIM1 protein is only increased in TILs, and not in cPBTs, compared to hPBTs. Furthermore, there was no difference in the protein levels of Orai1 among the three cell types. Notably, there was a discrepancy between the reduction in STIM1 transcripts and the increase in the STIM1 proteins observed in TILs compared to hPBTs. Similarly, recent work carried out on patient-derived mCRC cells showed that the mRNA encoding for STIM1 is equally expressed in primary and metastatic CRC cells, whereas the STIM1 protein was remarkably up-regulated in the latter [26]. Moreover, the Villalobos group reported that the Orai1 protein is significantly up-regulated in the HT29 CRC cell line compared to non-neoplastic cells, although there is no difference in its transcript expression levels between the two cell types [37]. The same study also clearly demonstrated that, although STIM2 is highly expressed at the transcript level, it is almost absent at the protein level in HT29 cells [37]. The protein expression of the multiple components of the Ca^2+^-handling machinery could be finely regulated by controlling mRNA stability [58]. For instance, SERCA2A mRNA presents a longer half-life in left ventricular myocytes with respect to stomach smooth muscle, thereby causing a 100-fold increase in SERCA2A protein expression in the former cells [59]. An alternative, but not mutually exclusive, hypothesis involves a higher rate of STIM1 protein production in TILs compared to hPBTs (and cPBTs) [60].

These findings suggested that SOCE up-regulation could also reflect differences in the regulatory mechanisms between cPBTs (and, possibly, TILs) and hPBTs. It has been recognised that enhanced DGK expression/activity in TILs attenuates tumour dismantling by limiting DAG-dependent pathways in favour of Ca^2+^-mediated signalling [45,46]. DGK phosphorylates DAG into phosphatidic acid, thereby interfering with the recruitment of DAG-dependent effectors, including PKC [45,46]. Notably, PKC may inhibit SOCE by phosphorylating Orai1 at NH_2_-terminal Ser-27 and Ser-30 residues [47,48]. Thus, we reasoned that the pharmacological blockade of DGK could selectively reduce SOCE by rescuing the PKC-mediated inhibition of Orai1 in TILs and cPBTs. Ten DGK isoforms have been detected and categorised into five subtypes in mammals: I (α, β, γ), II (δ, η, κ), III (ε), V (ζ and ι) and V (θ) [47,48]. Of these, type I DGKα may inhibit T-cell proliferation and function in the tumour microenvironment [45,61]. In agreement with our working hypothesis, R59949, which inhibits type I DGKs, and ritanserin, a selective blocker of DGKα, dramatically reduced CPA-evoked SOCE in both in vitro-expanded TILs and cPBTs, but not hPBTs. Conversely, blocking basal PKC activity with Gö-6983 strongly enhanced SOCE in hPBTs, but not in CTLs and TILs. These findings indicate that PKC exerts a weak inhibition of SOCE in TILs and cPBTs and that this is likely to reflect an increase in DGKα expression/activity that reduces DAG levels and DAG-dependent signalling. In agreement with this hypothesis, the direct stimulation of PKC with PMA decreased SOCE amplitude in all cell types. This finding shows that PKC can still inhibit SOCE in TILs and cPBTs, but the basal production of its physiological ligand, i.e., DAG, is maintained under the threshold to effectively reduce the SOCE amplitude. Altogether, our results strongly suggest that DGKα enables SOCE to be hyperactivated in TILs and cPBTs by preventing DAG-dependent PKC-mediated inhibition. Similarly, the PKC-dependent modulation of Orai1 activity has been reported in invasive melanoma cells [47] and in rat basophilic leukaemia cells [62].

SOCE supports the increase in [Ca^2+^]_i_ that arises in both CTLs and TILs following TCR stimulation at the immune synapse to support cancer cell killing [21,31,50,51]. Preliminary experiments revealed that the increase in SOCE led to a stronger oscillatory Ca^2+^ signal in TILs than in cPBTs, following stimulation with an anti-CD3 antibody, which is routinely employed to assess the strength of the specific T-cell response to TCR engagement [17,31,49]. However, when ex vivo-expanded TILs were stimulated with autologous mCRC cells, they displayed multiple Ca^2+^ signatures, the most frequent of which (i.e., “slow pattern” and “plateau pattern”, occurring in 53.8% of the cells) present a sustained increase in [Ca^2+^]_i_. The pharmacological blockade of SOCE with BTP-2 significantly increased the percentage of TILs displaying intracellular Ca^2+^ oscillations (from 46.08% to 100%) and significantly reduced the amount of total Ca^2+^ introduced into the cytosol during the immune response. Notably, pre-treating the TILs with BTP-2 improved their cytotoxic activity against the mCRC cells at each E:T ratio, and this increase became statistically relevant at a lower E:T ratio, suggesting that this effect was more evident when the TILs-mediated lysis was low. This feature is quite relevant, since, in non-haematological solid cancers, TILs often fail to kill target cells during 1:1 conjugation [63]. Our results are consistent with the recent report that a reduction in SOCE enhances CTL and natural killer cell (NK) cytotoxicity against cancer cells [21]. In accord, this and other studies showed that exaggerated Ca^2+^ entry in CTLs or NK can interfere with multiple steps of the immune response, e.g., migration and release of lytic granules, while it can enhance proliferation and apoptosis [21,52]. The mechanism(s) whereby exaggerated Ca^2+^ entry in TILs during mCRC cell killing reduces cytotoxic activity, e.g., increased apoptotic rate, is (are) under current investigation.

## 5. Conclusions

We provide the first evidence that SOCE is up-regulated in ex vivo-expanded TILs from mCRC patients. Extracellular Ca^2+^ entry is mainly enhanced by DGK activity, which prevents SOCE inhibition by PKC. The pharmacological blockade of SOCE with BTP-2 enhances TILs’ ability to kill autologous mCRC cells at low-density ratios. This preliminary evidence might suggest an alternative pharmacological strategy to improve the therapeutic outcome of ACT in mCRC patients.

## Figures and Tables

**Figure 1 cancers-14-03312-f001:**
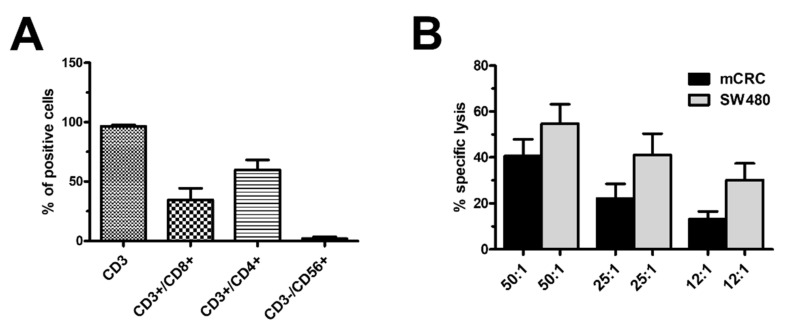
Characteristics of expanded TILs. Immunophenotype and potency were evaluated in TIL derived from 10 patients after 14 days’ expansion. (**A**) Analysis of surface antigen expression. Results are expressed as the mean and SD % receptor surface expression of CD3+, CD3+/CD4+, CD3+/CD8+ and CD3−/CD56+ cell populations. (**B**) Cytotoxic activity of TILs was tested in a 5-h cytotoxicity assay against the autologous mCRC cells (black column) and SW480 cell lines (grey column) at various E:T ratios ranging from 50:1 to 12:1. Means and SD of percentages of specific lysis are reported.

**Figure 2 cancers-14-03312-f002:**
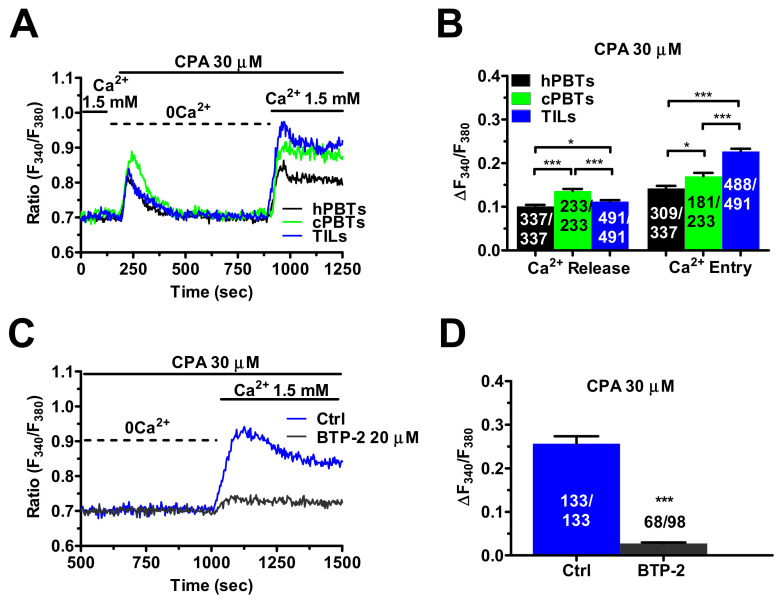
SOCE is enhanced in TILs. (**A**) The ER Ca^2+^ pool was depleted by challenging the cells with CPA (30 µM), a selective inhibitor of SERCA activity, in the absence of external Ca^2+^ (0Ca^2+^). Under 0Ca^2+^ conditions, CPA evoked a transient increase in [Ca^2+^]_i_ that was due to passive ER Ca^2+^ efflux through leakage channels, followed by Ca^2+^ sequestration by multiple mechanisms, such as mitochondria, Na^+^/Ca^2+^ exchanger (NCX) and plasma-membrane Ca^2+^-ATPase (PMCA) [26,37,38]. Subsequently, SOCE was evaluated by replenishing Ca^2+^ to the perfusate (1.5 mM). (**B**) Mean ± SE of the amplitude of CPA-induced intracellular Ca^2+^ release and CPA-induced SOCE in hPBTs, cPBTs and TILs. One-way ANOVA followed by the post hoc Bonferroni test: * *p* < 0.05 and *** *p* < 0.001. (**C**) Pre-incubating TILs with BTP-2 (20 µm, 20 min), a selective Orai1 inhibitor, strongly reduced SOCE. BTP-2 was added in the continuous presence of extracellular Ca^2+^ after the recovery to the baseline of CPA-evoked intracellular Ca^2+^ mobilisation. (**D**) Mean ± SE of SOCE amplitude in the absence (Ctrl) and presence of BTP-2. Student’s *t*-test: *** *p* < 0.001. In panels (**B**,**D**), the numbers within or above the histogram bars indicate the number of responding cells over the total number of analysed cells.

**Figure 3 cancers-14-03312-f003:**
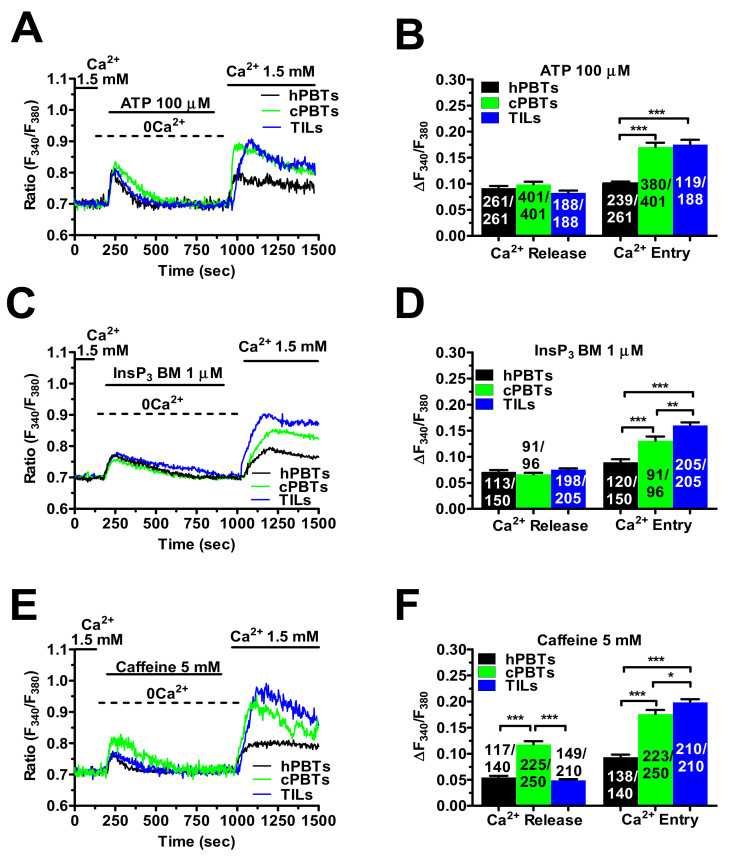
SOCE induced by ER Ca^2+^ depletion through InsP_3_Rs and RyRs is enhanced in TILs. (**A**) The ER Ca^2+^ pool was depleted by challenging the cells with the InsP_3_-producing autacoid, ATP (100 µM) in the absence of external Ca^2+^ (0Ca^2+^). Under 0Ca^2+^ conditions, ATP evoked a transient increase in [Ca^2+^]_i_ that rapidly returned to the baseline upon clearance of cytosolic Ca^2+^ by mitochondria, SERCA, NCX and PMCA [26,37,38]. Subsequently, SOCE was evaluated by replenishing Ca^2+^ (1.5 mM) to the perfusate in the absence of the agonist, which was removed 100 sec before re-addition of extracellular Ca^2+^. (**B**) Mean ± SE of the amplitude of ATP-induced intracellular Ca^2+^ release and SOCE in hPBTs, cPBTs and TILs. One-way ANOVA followed by the post hoc Bonferroni test: *** *p* < 0.001. (**C**) The ER Ca^2+^ pool was depleted by challenging the cells with InsP_3_BM (1 µM) in the absence of external Ca^2+^ (0Ca^2+^). Under 0Ca^2+^ conditions, InsP_3_BM evoked a transient increase in [Ca^2+^]_i_ that rapidly returned to the baseline upon clearance of cytosolic Ca^2+^ by mitochondria, SERCA, NCX and PMCA [38]. Subsequently, SOCE was evaluated by replenishing Ca^2+^ to the perfusate (1.5 mM) in the absence of the agonist, which was removed 100 sec before re-addition of extracellular Ca^2+^. (**D**) Mean ± SE of the amplitude of InsP_3_BM-induced intracellular Ca^2+^ release and SOCE in hPBTs, cPBTs and TILs. One-way ANOVA followed by the post hoc Bonferroni test: ** *p* < 0.01 and *** *p* < 0.001. (**E**) The ER Ca^2+^ pool was depleted by challenging the cells with caffeine (5 mM), a selective RyR agonist, in the absence of external Ca^2+^ (0Ca^2+^). Under 0Ca^2+^ conditions, caffeine evoked a transient increase in [Ca^2+^]_i_ that rapidly returned to the baseline upon clearance of cytosolic Ca^2+^ by mitochondria, SERCA, NCX and PMCA [40]. Subsequently, SOCE was evaluated by replenishing Ca^2+^ to the perfusate (1.5 mM) in the absence of the agonist, which was removed 100 sec before re-addition of extracellular Ca^2+^. (**F**) Mean ± SE of the amplitude of caffeine-induced intracellular Ca^2+^ release and SOCE in hPBTs, cPBTs and TILs. One-way ANOVA followed by the post hoc Bonferroni test: * *p* < 0.05 and *** *p* < 0.001. In panels (**B**,**D**,**F**), the numbers within or above the histogram bars indicate the number of responding cells over the total number of analysed cells.

**Figure 4 cancers-14-03312-f004:**
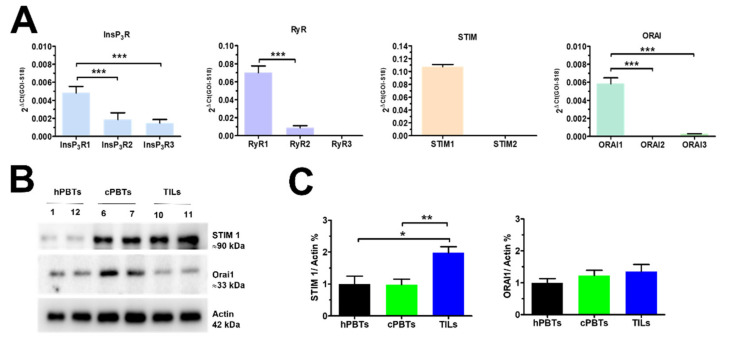
Molecular characterisation of the Ca^2+^-handling machinery in mCRC-derived T cells. (**A**) Transcriptomic profile of InsP_3_R, RyR, STIM, and Orai isoforms expressed in hPBTs. Data are calculated as 2^ΔCt(GOI-S18)^ and presented as mean ± SE from seven biological replicates from three donors for each condition (hPBTs, cPBTs and TILs). One-way ANOVA followed by the post hoc Bonferroni test: *** *p* < 0.001. (**B**) Expression of STIM1 and Orai1 proteins in hPBTs, cPBTs and TILs. Major bands of the expected molecular weight for each protein were observed. (**C**) Mean ± SE of the densitometric analysis of three different experiments, each carried out on a different donor. STIM1 and Orai1 protein bands were derived from the same experiment; thus, actin bands are common to the two proteins studied. One-way ANOVA followed by the post hoc Bonferroni test: * *p* < 0.05 and ** *p* < 0.01. Whole Western blot showing all bands in Appendix A.

**Figure 5 cancers-14-03312-f005:**
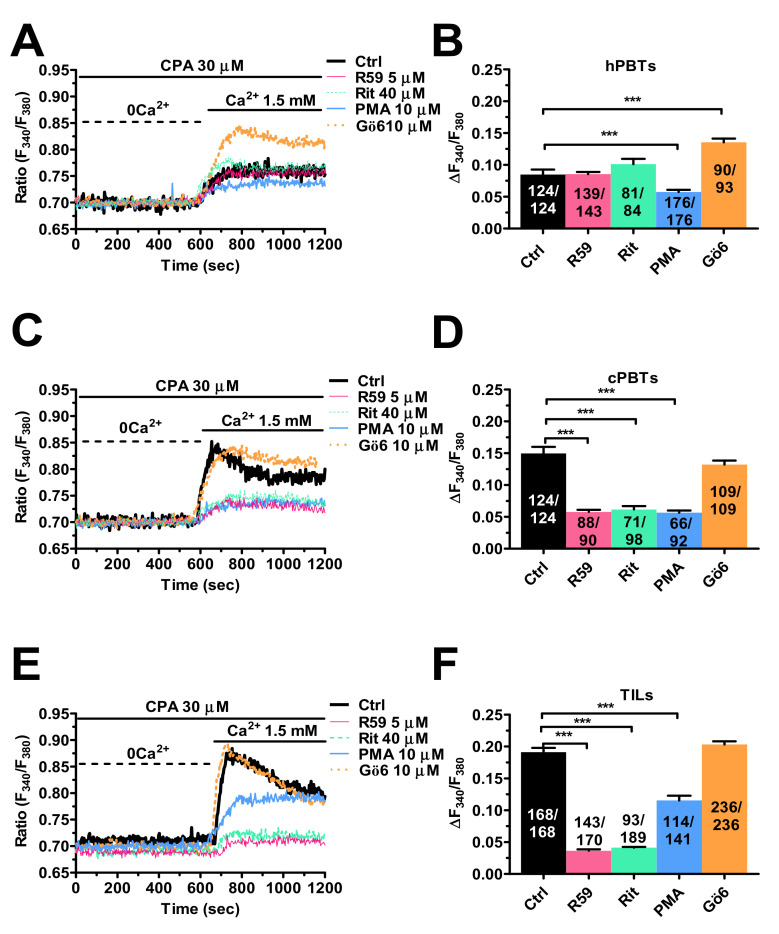
DGK regulates SOCE amplitude in TILs. (**A**) SOCE was activated by challenging hPBTs with CPA (30 µM) for 20 min under 0Ca^2+^ conditions (not shown) followed by restitution of extracellular Ca^2+^ to the perfusate (1.5 mM). SOCE was measured in the absence (Ctrl) and in the presence of the following drugs: ritanserin (40 µM, 10 min; Rit) or R59949 (5 µM, 10 min; R59), which inhibit DGK activity; Gö-6983 (10 µM, 10 min; Gö-6), which blocks PKC; and PMA (10 µM, 10 min), which stimulates PKC. CPA-evoked SOCE was reduced by PMA, enhanced by Gö-6983, and not affected by ritanserin or R59949. (**B**) Mean ± SE of SOCE amplitude in hPBTs under the designated treatments. One-way ANOVA followed by the post hoc Bonferroni test: *** *p* < 0.001. (**C**) SOCE was activated by challenging cPBTs with CPA (30 µM) for 20 min under 0Ca^2+^ conditions (not shown) followed by restitution of extracellular Ca^2+^ to the perfusate (1.5 mM). SOCE was measured in the absence (Ctrl) and in the presence of the following drugs: ritanserin (40 µM, 10 min; Rit) or R59949 (5 µM, 10 min, R59), which inhibit DGK activity; Gö-6983 (10 µM, 10 min; Gö-6), which blocks PKC; and PMA (10 µM, 10 min), which stimulates PKC. CPA-evoked SOCE was reduced by PMA, ritanserin or R59949 and not affected by Gö-6983. (**D**) Mean ± SE of SOCE amplitude in cPBTs under the designated treatments. One-way ANOVA followed by the post hoc Bonferroni test: *** *p* < 0.001. (**E**) SOCE was activated by challenging TILs with CPA (30 µM) for 20 min under 0Ca^2+^ conditions (not shown) followed by restitution of extracellular Ca^2+^ to the perfusate (1.5 mM). SOCE was measured in the absence (Ctrl) and in the presence of the following drugs: ritanserin (40 µM, 10 min; Rit) or R59949 (5 µM, 10 min; R59), which inhibit DGK activity; Gö-6983 (10 µM, 10 min; Gö-6), which blocks PKC; and PMA (10 µM, 10 min), which stimulates PKC. CPA-evoked SOCE was reduced by PMA, ritanserin or R59949 and not affected by Gö-6983. (**F**) Mean ± SE of SOCE amplitude under the designated treatments. One-way ANOVA followed by the post hoc Bonferroni test: *** *p* < 0.001. In panels (**B**,**D**,**F**), the numbers within or above the histogram bars indicate the number of responding cells over the total number of analysed cells.

**Figure 6 cancers-14-03312-f006:**
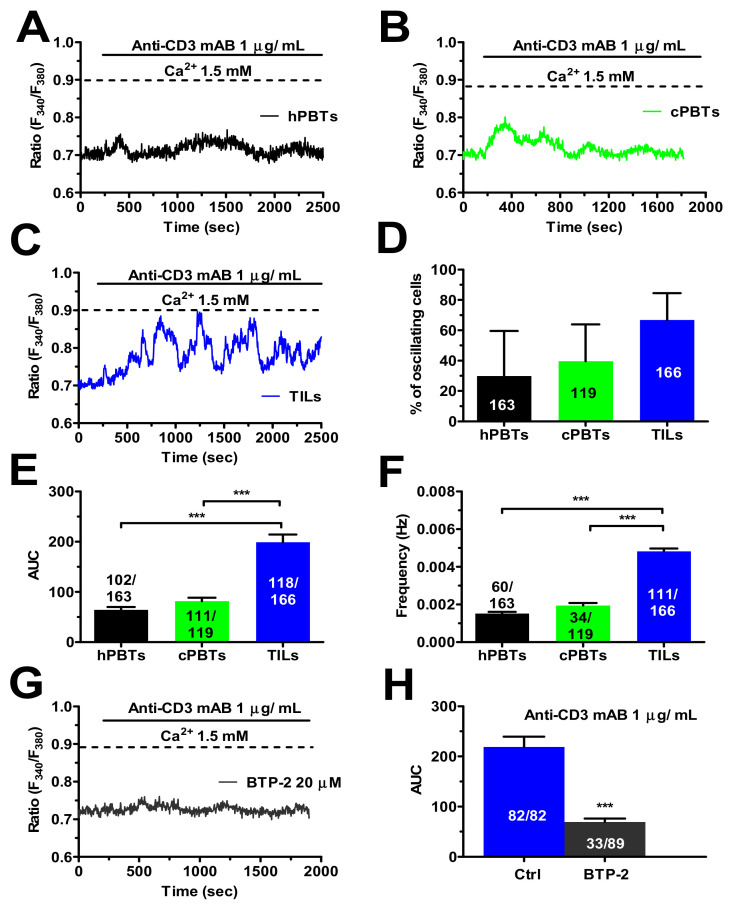
TCR stimulation by an anti-CD3 mAB evoked a stronger oscillatory Ca^2+^ response in TILs than in hPBTs and cPBTs. (**A**) Intracellular Ca^2+^ oscillations evoked in hPBTs by an anti-CD3 mAB (1 µg/mL). (**B**) Intracellular Ca^2+^ oscillations evoked in cPBTs by an anti-CD3 mAB (1 µg/mL). (**C**) Intracellular Ca^2+^ oscillations evoked in TILs by an anti-CD3 mAB (1 µg/mL). (**D**) Mean ± SE of the percentage of hPBTs, cPBTs and TILs responding to the anti-CD3 mAB stimulation. (**E**) Mean ± SE of the AUC of the spiking Ca^2+^ signal arising in the three cell types upon stimulation with the anti-CD3 mAB. One-way ANOVA followed by the post hoc Bonferroni test: *** *p* < 0.001. (**F**) Mean ± SE of oscillation frequency in the three cell types displaying the spiking Ca^2+^ response (i.e., more than 1 Ca^2+^ transient) to the anti-CD3 mAB. One-way ANOVA followed by the post hoc Bonferroni test: *** *p* < 0.001. (**G**) Pre-treating TILs with BTP-2 (20 µm, 20 min), which selectively inhibits SOCE by blocking extracellular Ca^2+^ influx through Orai1, dampens the intracellular Ca^2+^ oscillations evoked by the anti-CD3 mAB (1 µg/mL). (**H**) Mean ± SE of the AUC of the spiking Ca^2+^ signal arising in TILs challenged with the anti-CD3 mAB in the absence (Ctrl) and presence of BTP-2. Student’s *t*-test: *** *p* < 0.001. In panels (**B**,**D**,**F**,**H**), the numbers within or above the histogram bars indicate the number of responding cells over the total number of analysed cells.

**Figure 7 cancers-14-03312-f007:**
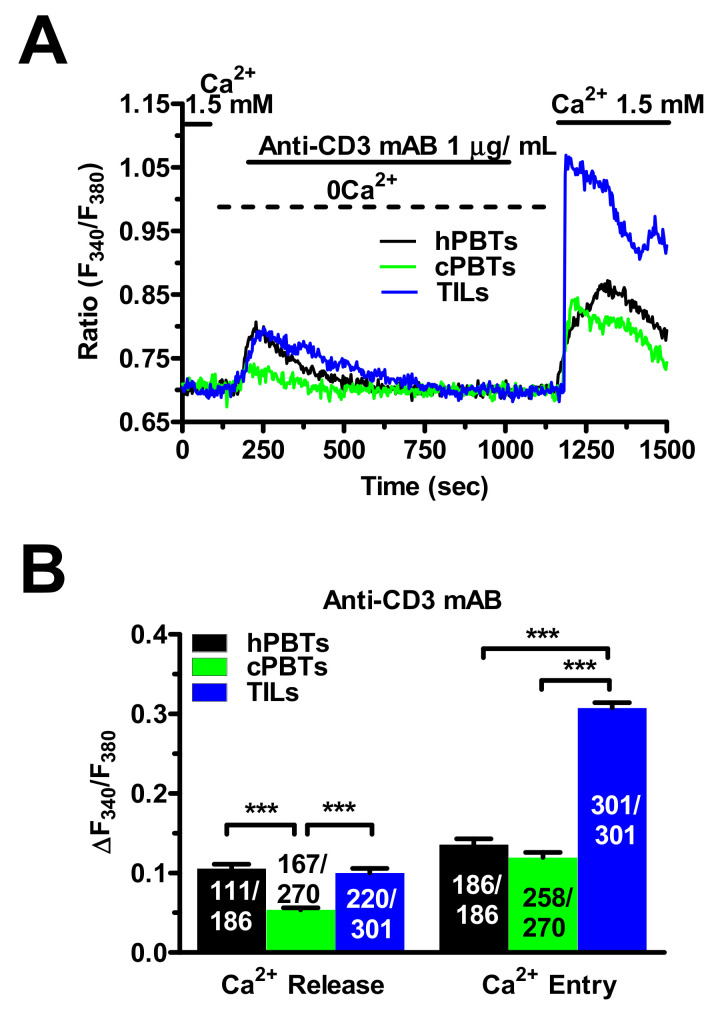
SOCE evoked by an anti-CD3 mAB is larger in TILs compared to hPBTs and cPBTs. (**A**) Intracellular Ca^2+^ mobilisation and SOCE activation in response to exogenous administration of an anti-CD3 mAB (1 µg/mL) in hPBTs, cPBTs and TILs. The anti-CD3 mAB was administered under 0Ca^2+^ conditions to evaluate endogenous Ca^2+^ release. Extracellular Ca^2+^ was returned to the perfusate (1.5 mM) after the recovery of the initiate Ca^2+^ transient to the baseline to measure SOCE. (**B**) Mean ± SE of the amplitude of ER Ca^2+^ mobilisation and SOCE evoked by the anti-CD3 mAB in the three cell types. One-way ANOVA followed by the post hoc Bonferroni test: *** *p* < 0.001. The numbers within or above the histogram bars indicate the number of responding cells over the total number of analysed cells.

**Figure 8 cancers-14-03312-f008:**
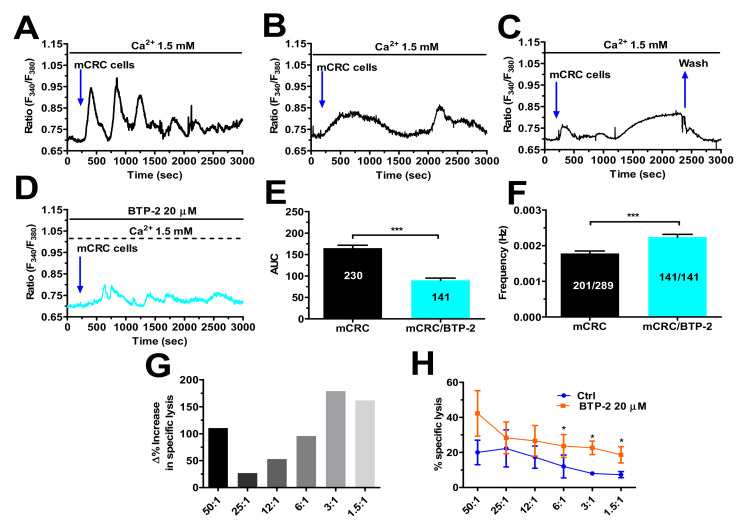
Reducing SOCE with BTP-2 increases TIL cytotoxicity against autologous mCRC cells. (**A**–**C**): Three distinct patterns of intracellular Ca^2+^ signals evoked in TILs plated on 8 mm glass coverslips (4 × 10^5^ cells/mL) by the contact with autologous mCRC cells (1 × 10^4^ cells/mL): (**A**), intracellular Ca^2+^ oscillations (“oscillatory pattern”); (**B**), two slow, intracellular Ca^2+^ waves (“slow pattern”); and (**C**), fast Ca^2+^ transient followed by a slow increase in [Ca^2+^]_i_ that ultimately achieved a plateau phase (“plateau pattern”). (**D**) Pre-treating TILs with BTP-2 (20 µm, 20 min) to selectively inhibit SOCE dampened the Ca^2+^ response to TCR engagement by autologous mCRC cells (1 × 10^4^ cells/mL) and suppressed the “slow” and “plateau” patterns. (**E**) Mean ± SE of the AUC of the spiking Ca^2+^ signal arising in TILs challenged with autologous mCRC cells in the absence (Ctrl) and presence of BTP-2. Student’s *t*-test: *** *p* < 0.001. (**F**) Mean ± SE of oscillation frequency in TILs exposed to autologous mCRC cells in the absence (Ctrl) and presence of BTP-2. Student’s *t*-test: *** *p* < 0.001. (**G**) Δ of the % increase in specific lysis at each E:T ratio. (**H**) Cytotoxic activity of TIL derived from five patients was tested through a 5 h cytotoxicity assay against autologous mCRC cells in medium alone (Ctrl) or in the presence of BTP-2 at various E:T ratios. Data are presented as mean ± SE from three separate experiments. Student’s *t*-test: * *p* < 0.05. In panels (**E**,**F**), the numbers within or above the histogram bars indicate the number of responding cells over the total number of analysed cells.

## Data Availability

Data supporting the reported results can be obtained upon reasonable request to the authors.

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
