# Peer review of "Store-Operated Ca2+ Entry Is Up-Regulated in Tumour-Infiltrating Lymphocytes from Metastatic Colorectal Cancer Patients"

_cancers, 2022, doi:10.3390/cancers14143312_

Round 1

Reviewer 1 Report

In this study, Faris and al. are suggesting that tumour infiltrating lymphocytes expanded in vitro show an increases in store-operated calcium entry compared to peripheral blood T cells from either the same cancer patient or healthy donors. This effect could ne explained by increased activity of diacylglycerol kinase which prevent protein kinase C dependent inhibition of ORAI1. This study suggest that inhibition of SOCE in TIL expanded in vitro could improve adoptive cell therapy of patients suffering from colorectal carcinoma metastatic disease. However, this claim would need to be verified experimentally.

Overall the manuscript would benefit from the addition of more details in the figures and figure legends. My main concerns is the analysis of the RT-qPCR data which I believe were not done properly.

Major concerns:

Supplementary Figure 3 and Figure 4a: Please explain how the RT-qPCR data were analysed. From the materials and methods section, it is stated that qPCR data are presented as dCT of Gene/S18. Correct calculation for RT-qPCR analysis using the comparative method is dCT = CTgene – CT18s. Please refer to RT-qPCR protocols ( eg. DOI: 10.1002/0471142727.mb1508s73) and reanalyse all RT-qPCR data. The authors are suggesting that there is a STIM1 mRNA decreases in TIL but an increase of STIM1 protein. If after reanalysing the qPCR data it is still the case, the authors need to comments on this in the discussion.

Line 99: depletion of SOCE prevent tumour engraftment and promote tumour growth (ref 23). Please check again this reference. I believe they are suggesting that SOCE in CTL is essential to prevent tumour engraftment and that tumour are forming when STIM is depleted from SOCE.

Are the TIL cells used throughout the study are derived from the same patients? Please be more specific.

Figure 1: Flow cytometry figures are too small and of poor resolution. Figure 1a: show the flow cytometry data plot either in figure 1 or supplementary figure, not just the % in a bar graph. Also indicate number of cells/independent experiments in figure legend. Figure legend of Fig 1B match fig 1D. Please review and correct the figure legend. Please better describe Figure 1 in the main text. In Figure 1A, are the cells derived from one patient or it is a summary of multiple patients? Better describe the cytotoxic assay used in D.

Figure 2: Figure 2B and D: What does the numbers (eg. 337/337) represents? Please explain in figure legends ( and all other figures where they are present). Why is the calcium release by CPA not included in the calcium traces of figure 2c?Please add a line on graph A and B ( also Fig3) representing calcium add back and concentration of Ca2+ added.  Also add Ca2+ addition concentration in the figure legends ( also for Fig 3).

Figure 3: Have the authors investigated the effect of ATP in presence of Ca2+? It would be interesting to investigate if there is a more sustain Ca2+ entry in the TILs compared to hPBTs?

Figure 4a: Please indicates in Figure legend how relative expression was calculated. Is it dCt as in supplementary Figure 3?

Figure 4B: Add molecular weight to the western blot. Please explain in figure legend what numbers above western blot means (eg. hPBTS 1, 12)

Figure 5: Is the CPA calcium release from the ER affected by the multiples drugs used in Figure 5?

Figure 8: This figure is labelled as Figure 7 in the Figure legend.  

Overall comments regarding Figures: Figures are too small and text on figures is difficult to read. Figure legends should contains more experimental information for the reader to understand the data easily.

Minor errors:

Supp 3: decrease in mRNA STIM1 and Figure 4: increase in sTIM1 protein

Line 28: cytotoxic t lymphocyte: capitalise the T

Line 64, 395, 401, 408: formatting error in front of the 80%?

Line 70: ex vivo rapidly expanded-

Line 74: which are ex vivo induced to recognize

Line 124:  trypan blue die replace by trypan blue dye

Please give catalogue numbers of antibodies used for FACS

Line 161: I don’t understand the unit measure of 20 µl Ci

Line 185: 10 (for STIM1) 20 µg (for ORAI1): please change for : 10 µg (STIM1) and …

Line 195: add mM to each

Supplementary Table 2: typo in ORAI1 protein name.

Line 239: Add + next to CD3/

Line 335: STIM. Should it be STIM1?

Author Response

Dear Reviewer #1,

We are gratefully thankful for your comments on our manuscript entitled: “Store-operated Ca2+ entry is up-regulated in tumour infiltrating lymphocytes from metastatic colorectal cancer patients” for publication as Research Article in Cancers – Special Issue Advances in Cancer Immunotherapy.

We amended the manuscript according to your suggestions and strongly believe that your comments remarkably improve the quality of our work.

Major concerns

Supplementary Figure 3 and Figure 4a: Please explain how the RT-qPCR data were analysed. From the materials and methods section, it is stated that qPCR data are presented as dCT of Gene/S18. Correct calculation for RT-qPCR analysis using the comparative method is dCT = CTgene – CT18s. Please refer to RT-qPCR protocols (eg. DOI: 10.1002/0471142727.mb1508s73) and reanalyse all RT-qPCR data. The authors are suggesting that there is a STIM1 mRNA decreases in TIL but an increase of STIM1 protein. If after reanalysing the qPCR data it is still the case, the authors need to comment on this in the discussion.

We apologize for not being clear in the description of the method we have used to quantified qPCR data. It is a variant of ddCt method, which, however, allows to express the mRNA quantities of GOI relative to internal control (S18). The values are calculated using formula 2^-(CTgene – CTs18), and plotted as Mean ± SD.

In terms of fold change of the experimental sample to CTR the results are exactly the same as if calculated using ddCT method, because, mathematically 2^-((CT(GOIsample) – CT(S18sample)) - (CT(GOIcontrol) – CT(S18control))) = 2^-(CT(GOIsample) – CT(S18sample)) ÷ 2^-(CT(GOIcontrol) – CT(S18control)).

However, this method has a number of advantages compared to classical ddCT method. It shows the fold change of GOI relative to S18 internal control, which allows to appreciate the level of GOI’s mRNA expression compared to S18, which gives an immediate information about the level of GOI’s the mRNA expression. The CTR sample is not valued to 1 or 100%, which allows to perform statistical analysis (ANOVA) considering the internal variability of all samples.

In the revised version of the manuscript, we have described better qPCR quantification in the Methods section. The data were reanalyzed and we confirm the results presented in the original version of the manuscript. The discrepancy between STIM1 mRNA and protein data has been discussed in lines 616-631.

Line 99: depletion of SOCE prevent tumour engraftment and promote tumour growth (ref 23). Please check again this reference. I believe they are suggesting that SOCE in CTL is essential to prevent tumour engraftment and that tumour are forming when STIM is depleted from SOCE.

The Referee is fully right. Ref. 23 clearly shows that SOCE inhibition favours (not prevents) the engraftment of cancer cells in a mouse model of melanoma. The text has been amended accordingly “Thereby preventing the engraftment” has been replaced with “thereby favouring the engraftment”. We thank the Referee for this observation.

Are the TIL cells used throughout the study derived from the same patients? Please be more specific.

We thank the Referee for this observation. This information has been provided in Section 2.9, Statistics. We stated that: “All the Ca2+ imaging data have been collected from T-cells deriving from three donors. Tumour-derived T-cells, i.e., cPBTs and TILs, isolated from all the 10 patients have been used throughout the study. Therefore, cPBTs and TILs expanded from three distinct donors could be used in different experimental conditions. Likewise, hPBTs isolated from three healthy donors were used for Ca2+ imaging experiments”.

Figure 1: Flow cytometry figures are too small and of poor resolution. Figure 1a: show the flow cytometry data plot either in figure 1 or supplementary figure, not just the % in a bar graph. Also indicate number of cells/independent experiments in figure legend. Figure legend of Fig 1B match fig 1D. Please review and correct the figure legend. Please better describe Figure 1 in the main text. In Figure 1A, are the cells derived from one patient or it is a summary of multiple patients? Better describe the cytotoxic assay used in D.

We thank the Referee for these observations. In the text and in the figure legend, we have specified that results derived from the 10 patients enrolled in this study. The cytotoxicity assay was better described in the Material and Methods section (lines 169-177) and in the Results section (lines 259-267). The figure legend has been clarified. We improved the resolution and the of the flow cytometry figures, the data plot is now available as a supplementary figure. The figure legend has been improved.

Figure 2: Figure 2B and D: What does the numbers (eg. 337/337) represents? Please explain in figure legends (and all other figures where they are present). Why is the calcium release by CPA not included in the calcium traces of figure 2c? Please add a line on graph A and B (also Fig3) representing calcium add back and concentration of Ca2+ added.  Also add Ca2+ addition concentration in the figure legends (also for Fig 3).

We thank the Referee for these observations. The numbers indicate the number of responsive cells over the total number of cells analyzed. This information has now been provided in all the relevant Figure legends. CPA-evoked Ca2+ release in Figure 2C has not been shown since we focused on the effect of BTP-2 on SOCE and BTP-2 has been added after Ca2+ release. We implemented the description of the protocol in Figure 2c. Furthermore, we have changed all the figures and added a dashed line to indicate the time of Ca2+ readdition. Finally, we have indicated the concentration of external Ca2+ both in the figures and in the legends.

Figure 3: Have the authors investigated the effect of ATP in presence of Ca2+? It would be interesting to investigate if there is a more sustain Ca2+ entry in the TILs compared to hPBTs?

We thank the referee for this suggestion. We were able to carry out these experiments upon your request. In addition, we have also measured the Ca2+ response to CPA in the presence of extracellular Ca2+. The corresponding results are now shown in Figure S3 (CPA) and in Figure S5 (ATP) and confirm the results of the Ca2+ add-back protocol.

Figure 4a: Please indicates in Figure legend how relative expression was calculated. Is it dCt as in supplementary Figure 3?

We thank the referee for this comment. Relative expression has been calculated as 2ΔCt(GOI-S18), as now also explained in Figure legend.

Figure 4B: Add molecular weight to the western blot. Please explain in figure legend what numbers above western blot means (eg. hPBTS 1, 12).

We thank the referee for these comments. Molecular weights have been added to the western blot. The number above the western blots refer to sample numbers and were deleted from the revised manuscript.

Figure 5: Is the CPA calcium release from the ER affected by the multiple drugs used in Figure 5?

We thank the referee for these comments. We focused on SOCE and, therefore, did not show ER Ca2+ release. Moreover, all the drugs have been added to the perfusate after CPA-evoked intracellular Ca2+ transient. This information has been provided in the revised manuscript.

Figure 8: This figure is labelled as Figure 7 in the Figure legend.

We thank the referee for this observation. Figure 8 has been correctly labelled in the Figure legend.

Overall comments regarding Figures: Figures are too small and text on figures is difficult to read. Figure legends should contain more experimental information for the reader to understand the data easily.

We thank the Referee for this suggestion. All the fonts in the figures were increased and the figures enlarged. We modified all figure legends to add more experimental information and help the reader, as you requested.

Minor errors:

Supp 3: decrease in mRNA STIM1 and Figure 4: increase in STIM1 protein

Analysis of the RT-PCR data confirmed that STIM1 transcripts are down-regulated, while STIM1 protein is up-regulated. This discrepancy has been discussed in lines 616-631.

Line 28: cytotoxic t lymphocyte: capitalise the T

The T has been capitalised. We thank the Referee for this observation.

Line 64, 395, 401, 408: formatting error in front of the 80%?

All the errors have been formatted. We thank the Referee for these observations.

Line 70: ex vivo rapidly expanded

Rapidly has been removed.

Line 74: which are ex vivo induced to recognize

Ex vivo has been removed

Line 124:  trypan blue die replace by trypan blue dye

“With” has been replaced by “by”.

Please give catalogue numbers of antibodies used for FACS

The catalogue number of each antibody has been indicated.

Line 161: I don’t understand the unit measure of 20 µl Ci

We apologise for this typo that has been removed.

Line 185: 10 (for STIM1) 20 µg (for ORAI1): please change for: 10 µg (STIM1) and …

The text has been reworded according to your right indications.

Line 195: add mM to each

mM has been added to each concentration.

Supplementary Table 2: typo in ORAI1 protein name.

The typo in ORAI1 protein name has been corrected. Thank you!

Line 239: Add + next to CD3/

+ has been added next to CD3/.

Line 335: STIM. Should it be STIM1?

Yes, definitively. We thank the Referee for noticing this (dozens of manuscript readings are never enough!).

Once again, we thank you for the careful evaluation of our manuscript and do hope that you will now regard is suitable for publication on Cancers.

Sincerely,

Francesco Moccia

Francesco Moccia, PhD

Laboratory of General Physiology,

Department of Biology and Biotechnology “L. Spallanzani”

University of Pavia,

Via Forlanini 6, 27100, Pavia, Italy.

Tel: 0039 0382 987614.

Fax: 0039 0382 987527.

Dear Reviewer #1,

We are gratefully thankful for your comments on our manuscript entitled: “Store-operated Ca2+ entry is up-regulated in tumour infiltrating lymphocytes from metastatic colorectal cancer patients” for publication as Research Article in Cancers – Special Issue Advances in Cancer Immunotherapy.

We amended the manuscript according to your suggestions and strongly believe that your comments remarkably improve the quality of our work.

Major concerns

Supplementary Figure 3 and Figure 4a: Please explain how the RT-qPCR data were analysed. From the materials and methods section, it is stated that qPCR data are presented as dCT of Gene/S18. Correct calculation for RT-qPCR analysis using the comparative method is dCT = CTgene – CT18s. Please refer to RT-qPCR protocols (eg. DOI: 10.1002/0471142727.mb1508s73) and reanalyse all RT-qPCR data. The authors are suggesting that there is a STIM1 mRNA decreases in TIL but an increase of STIM1 protein. If after reanalysing the qPCR data it is still the case, the authors need to comment on this in the discussion.

We apologize for not being clear in the description of the method we have used to quantified qPCR data. It is a variant of ddCt method, which, however, allows to express the mRNA quantities of GOI relative to internal control (S18). The values are calculated using formula 2^-(CTgene – CTs18), and plotted as Mean ± SD.

In terms of fold change of the experimental sample to CTR the results are exactly the same as if calculated using ddCT method, because, mathematically 2^-((CT(GOIsample) – CT(S18sample)) - (CT(GOIcontrol) – CT(S18control))) = 2^-(CT(GOIsample) – CT(S18sample)) ÷ 2^-(CT(GOIcontrol) – CT(S18control)).

However, this method has a number of advantages compared to classical ddCT method. It shows the fold change of GOI relative to S18 internal control, which allows to appreciate the level of GOI’s mRNA expression compared to S18, which gives an immediate information about the level of GOI’s the mRNA expression. The CTR sample is not valued to 1 or 100%, which allows to perform statistical analysis (ANOVA) considering the internal variability of all samples.

In the revised version of the manuscript, we have described better qPCR quantification in the Methods section. The data were reanalyzed and we confirm the results presented in the original version of the manuscript. The discrepancy between STIM1 mRNA and protein data has been discussed in lines 616-631.

Line 99: depletion of SOCE prevent tumour engraftment and promote tumour growth (ref 23). Please check again this reference. I believe they are suggesting that SOCE in CTL is essential to prevent tumour engraftment and that tumour are forming when STIM is depleted from SOCE. The Referee is fully right. Ref. 23 clearly shows that SOCE inhibition favours (not prevents) the engraftment of cancer cells in a mouse model of melanoma. The text has been amended accordingly “Thereby preventing the engraftment” has been replaced with “thereby favouring the engraftment”. We thank the Referee for this observation.

Are the TIL cells used throughout the study derived from the same patients? Please be more specific.

We thank the Referee for this observation. This information has been provided in Section 2.9, Statistics. We stated that: “All the Ca2+ imaging data have been collected from T-cells deriving from three donors. Tumour-derived T-cells, i.e., cPBTs and TILs, isolated from all the 10 patients have been used throughout the study. Therefore, cPBTs and TILs expanded from three distinct donors could be used in different experimental conditions. Likewise, hPBTs isolated from three healthy donors were used for Ca2+ imaging experiments”.

Figure 1: Flow cytometry figures are too small and of poor resolution. Figure 1a: show the flow cytometry data plot either in figure 1 or supplementary figure, not just the % in a bar graph. Also indicate number of cells/independent experiments in figure legend. Figure legend of Fig 1B match fig 1D. Please review and correct the figure legend. Please better describe Figure 1 in the main text. In Figure 1A, are the cells derived from one patient or it is a summary of multiple patients? Better describe the cytotoxic assay used in D.

We thank the Referee for these observations. In the text and in the figure legend, we have specified that results derived from the 10 patients enrolled in this study. The cytotoxicity assay was better described in the Material and Methods section (lines 169-177) and in the Results section (lines 259-267). The figure legend has been clarified. We improved the resolution and the of the flow cytometry figures, the data plot is now available as a supplementary figure. The figure legend has been improved.

Figure 2: Figure 2B and D: What does the numbers (eg. 337/337) represents? Please explain in figure legends (and all other figures where they are present). Why is the calcium release by CPA not included in the calcium traces of figure 2c? Please add a line on graph A and B (also Fig3) representing calcium add back and concentration of Ca2+ added.  Also add Ca2+ addition concentration in the figure legends (also for Fig 3).

We thank the Referee for these observations. The numbers indicate the number of responsive cells over the total number of cells analyzed. This information has now been provided in all the relevant Figure legends. CPA-evoked Ca2+ release in Figure 2C has not been shown since we focused on the effect of BTP-2 on SOCE and BTP-2 has been added after Ca2+ release. We implemented the description of the protocol in Figure 2c. Furthermore, we have changed all the figures and added a dashed line to indicate the time of Ca2+ readdition. Finally, we have indicated the concentration of external Ca2+ both in the figures and in the legends.

Figure 3: Have the authors investigated the effect of ATP in presence of Ca2+? It would be interesting to investigate if there is a more sustain Ca2+ entry in the TILs compared to hPBTs?

We thank the referee for this suggestion. We were able to carry out these experiments upon your request. In addition, we have also measured the Ca2+ response to CPA in the presence of extracellular Ca2+. The corresponding results are now shown in Figure S3 (CPA) and in Figure S5 (ATP) and confirm the results of the Ca2+ add-back protocol.

Figure 4a: Please indicates in Figure legend how relative expression was calculated. Is it dCt as in supplementary Figure 3?

We thank the referee for this comment. Relative expression has been calculated as 2ΔCt(GOI-S18), as now also explained in Figure legend.

Figure 4B: Add molecular weight to the western blot. Please explain in figure legend what numbers above western blot means (eg. hPBTS 1, 12).

We thank the referee for these comments. Molecular weights have been added to the western blot. The number above the western blots refer to sample numbers and were deleted from the revised manuscript.

Figure 5: Is the CPA calcium release from the ER affected by the multiple drugs used in Figure 5?

We thank the referee for these comments. We focused on SOCE and, therefore, did not show ER Ca2+ release. Moreover, all the drugs have been added to the perfusate after CPA-evoked intracellular Ca2+ transient. This information has been provided in the revised manuscript.

Figure 8: This figure is labelled as Figure 7 in the Figure legend.

We thank the referee for this observation. Figure 8 has been correctly labelled in the Figure legend.

Overall comments regarding Figures: Figures are too small and text on figures is difficult to read. Figure legends should contain more experimental information for the reader to understand the data easily.

We thank the Referee for this suggestion. All the fonts in the figures were increased and the figures enlarged. We modified all figure legends to add more experimental information and help the reader, as you requested.

Minor errors:

Supp 3: decrease in mRNA STIM1 and Figure 4: increase in STIM1 protein

Analysis of the RT-PCR data confirmed that STIM1 transcripts are down-regulated, while STIM1 protein is up-regulated. This discrepancy has been discussed in lines 616-631.

Line 28: cytotoxic t lymphocyte: capitalise the T

The T has been capitalised. We thank the Referee for this observation.

Line 64, 395, 401, 408: formatting error in front of the 80%?

All the errors have been formatted. We thank the Referee for these observations.

Line 70: ex vivo rapidly expanded

Rapidly has been removed.

Line 74: which are ex vivo induced to recognize

Ex vivo has been removed

Line 124:  trypan blue die replace by trypan blue dye

“With” has been replaced by “by”.

Please give catalogue numbers of antibodies used for FACS

The catalogue number of each antibody has been indicated.

Line 161: I don’t understand the unit measure of 20 µl Ci

We apologise for this typo that has been removed.

Line 185: 10 (for STIM1) 20 µg (for ORAI1): please change for: 10 µg (STIM1) and …

The text has been reworded according to your right indications.

Line 195: add mM to each

mM has been added to each concentration.

Supplementary Table 2: typo in ORAI1 protein name.

The typo in ORAI1 protein name has been corrected. Thank you!

Line 239: Add + next to CD3/

+ has been added next to CD3/.

Line 335: STIM. Should it be STIM1?

Yes, definitively. We thank the Referee for noticing this (dozens of manuscript readings are never enough!).

Once again, we thank you for the careful evaluation of our manuscript and do hope that you will now regard is suitable for publication on Cancers.

Sincerely,

Francesco Moccia

Francesco Moccia, PhD

Laboratory of General Physiology,

Department of Biology and Biotechnology “L. Spallanzani”

University of Pavia,

Via Forlanini 6, 27100, Pavia, Italy.

Tel: 0039 0382 987614.

Fax: 0039 0382 987527.

Reviewer 2 Report

The manuscript titled “Store-operated Ca2+ entry is up-regulated in tumour infiltrating lymphocytes from metastatic colorectal cancer patients” describes mCRC patients-derived ex vivo experiments to detect the relationship between extracellular Ca2+ entry and STIM1 protein via PKC related signaling. Overall, the manuscript is well-written and well prepared. There is some detailed information that needs to be clarified. The followings are some concerns and comments have been pointed out that the authors may want to consider.

1) About 18% references from list authors. Self-citation seems too high. Please remove unnecessary references.

2) Line 34: Please consistent “2+” as a superscript.

3) Line 40: The statement “…could improve their therapeutic outcome” is not accurate. The ex vivo experiments could not exact present real-world clinical treatment.

4) Line 106 Methods section: Please include CAT# and a detailed source for all the reagents used in this study to make your work reproducible relatively easier by other researchers.

5) Line 108 Patients and donors: In Figure 1, the authors stated TIL was derived from two different patients. Line 224, T-cells deriving from at least three donors. The sample size is important. Please specify how many patients’ samples have been tested in each figure, hPBTs, cPBTs, and TILs.

6) Line 144: Please remove extra self-citation.

7) Line 148: I’d suggest the authors briefly describe the background of why 14 days.

8) Line 226: Please use italic p as it refers to a p-value throughout the manuscript.

9) Line 240: “CD3neg” please consistent the format throughout the manuscript.

10) Line 264: I’d suggest the authors consistent “>>” or “>”. Please specify if there are some special meanings.

11) Line 358 Figure 4: Figure 4B and 4C, please confirm only two selected samples have been used in each group. Are there any more patients’ samples that have been used for western blotting to confirm the results?

12) Line 395 and line 408: The sign between “mean” and “SE” should be a “plus-minus sign”.

13) Line 424 Figure 6 and all other figures: I’d suggest the authors add arrows to the images to indicate when the reagents were added into the cells that should be clearer.

14) Line 488: Figure 7D, what’s the meaning of “mCRC cellsl”? It should be an extra “l”.

15) Line 611: Did the authors perform some experiments that genetically over-expression of STIM1 protein to support the conclusion?

16) Are there any insights in vitro experiments have been done to release the mechanisms?  For example, knockdown or genetically overexpress STIM1 or other genes.

Author Response

Dear Reviewer #2,

We are gratefully thankful for your comments on our manuscript entitled: “Store-operated Ca2+ entry is up-regulated in tumour infiltrating lymphocytes from metastatic colorectal cancer patients” for publication as Research Article in Cancers – Special Issue Advances in Cancer Immunotherapy.

We amended the manuscript according to your suggestions and strongly believe that your comments remarkably improve the quality of our work.

More specifically:

  • About 18% references from list authors. Self-citation seems too high. Please remove unnecessary references.

We thank the Referee for this comment. We truly believe we quoted all the literature that was relevant to the manuscript. We did our best to identify the references from our groups that the Referee could judge as unnecessary.

  • Line 34: Please consistent “2+” as a superscript.

We thank the Referee for this right observation. “2+” has been formatted as superscript.

  • Line 40: The statement “…could improve their therapeutic outcome” is not accurate. The ex vivo experiments could not exact present real-world clinical treatment.

We do thank the Referee for this suggestion. The sentence has been reworded in: “could improve their cytotoxic activity against cancer cells”.

4) Line 106 Methods section: Please include CAT# and a detailed source for all the reagents used in this study to make your work reproducible relatively easier by other researchers.

We thank the Referee for this comment. All catalogues number were included, whereas all the sources have already been detailed.

5) Line 108 Patients and donors: In Figure 1, the authors stated TIL was derived from two different patients. Line 224, T-cells deriving from at least three donors. The sample size is important. Please specify how many patients’ samples have been tested in each figure, hPBTs, cPBTs, and TILs.

The Referee is right. In the Figure 1 and in the text, we have better specified that results of patients’ immunophenotype and potency refer to the 10 patients enrolled in the study. We have also shown dot plots of TILs derived from two representative patients in which we have documented a preferential expansion of CD3+/CD4+ or CD3+/CD8+ cell populations. As illustrated in section 2.9 Statistics, in each figure, we have tested T-cells (hPBTs, cPBTs and TILs) isolated from three different donors. All this information has now been made clearer.

6) Line 144: Please remove extra self-citation.

Extra self-citations have been removed.

7) Line 148: I’d suggest the authors briefly describe the background of why 14 days.

We thank the Referee for this suggestion. The background of why 14 days are required to expand CD45 positive cells has now been described.

8) Line 226: Please use italic p as it refers to a p-value throughout the manuscript.

We thank the Referee for this suggestion. “P” has been replaced with “p” to be consistent with the remainder of the manuscript.

9) Line 240: “CD3neg” please consistent the format throughout the manuscript.

We thank the Referee for this suggestion. “CD3neg” has been replaced with “CD3-“.

10) Line 264: I’d suggest the authors consistent “>>” or “>”. Please specify if there are some special meanings.

We thank the Referee for this suggestion. “>>” has been replaced with “>“.

11) Line 358 Figure 4: Figure 4B and 4C, please confirm only two selected samples have been used in each group. Are there any more patients’ samples that have been used for western blotting to confirm the results?

We confirm that, in Figure 4B, two selected samples from two distinct donor for each condition are shown. The total number of analyzed donors for each condition is three and is comprised in the densitometric analysis in Figure 4C. This information has been made clearer in the text.

12) Line 395 and line 408: The sign between “mean” and “SE” should be a “plus-minus sign”.

We thank the Referee for this suggestion. These typos have been carefully amended.

13) Line 424 Figure 6 and all other figures: I’d suggest the authors add arrows to the images to indicate when the reagents were added into the cells that should be clearer.

In all our papers, we always indicate the time of exposure to the reagents by placing black bars above the Ca2+ tracings. This strategy does not only indicate when reagents are added, but also when they are withdrawn from the perfusate. This approach is also the most currently used by our colleagues worldwide and is the most appropriate to exactly show the timings of addition of each drug. We would like to maintain this style also in the present publication, if possible.

14) Line 488: Figure 7D, what’s the meaning of “mCRC cellsl”? It should be an extra “l”.

We thank the Referee for this observation. This typo has been amended.

15) Line 611: Did the authors perform some experiments that genetically over-expression of STIM1 protein to support the conclusion?

We thank the Referee for this observation. We did not assess this issue through the genetic manipulation of STIM1. This assertion seems to be reasonable since STIM1 regulates SOCE in human T-cells. However, in order to address your comment, we softened our conclusion by rephrasing the sentence as follows: “Extracellular Ca2+ entry is mainly enhanced by DGK activity”.

16) Are there any insights in vitro experiments have been done to release the mechanisms?  For example, knockdown or genetically overexpress STIM1 or other genes.

We thank the Referee for this observation. We did genetically knockdown STIM1 or ORAI1 since our data regarding their gene/protein expression indicated that the increase in SOCE was mainly due to posttranslational modifications, such as those depending on DGK activity. Nevertheless, future work could be devoted to address this comment, as now stated in line 683.

Once again, we do thank you for the careful evaluation of the manuscript. We do hope that you will now regard our manuscript suitable for publication on Cancers.

Sincerely,

Francesco Moccia

Francesco Moccia, PhD

Laboratory of General Physiology,

Department of Biology and Biotechnology “L. Spallanzani”

University of Pavia,

Via Forlanini 6, 27100, Pavia, Italy.

Tel: 0039 0382 987614.

Fax: 0039 0382 987527.

Round 2

Reviewer 1 Report

There are two minor points that the author did not addressed correctly:

1) Author did not understand the following comment” Line 124: trypan blue die replace by trypan blue dye. Now line 125: “die” should be replaced by “dye”.

 2)Lane 170: 20 µl Ci. I still don’t understand what unit measure “ µl Ci” represent.

Author Response

Dear Reviewer #1,

We are gratefully thankful for your comments on our manuscript entitled: “Store-operated Ca2+ entry is up-regulated in tumour infiltrating lymphocytes from metastatic colorectal cancer patients” for publication as Research Article in Cancers – Special Issue Advances in Cancer Immunotherapy.

We amended the manuscript according to your suggestions and strongly believe that your comments remarkably improve the quality of our work.

More specifically:

1) Author did not understand the following comment” Line 124: trypan blue die replace by trypan blue dye. Now line 125: “die” should be replaced by “dye”.

We replaced “die” we “dye. Thanks again for noticing this typo.

2) Lane 170: 20 µl Ci. I still don’t understand what unit measure “µl Ci” represent.

The term Ci is a typo that has been removed. Thanks again for noticing this typo.

Once again, we thank you for the careful evaluation of our manuscript and do hope that you will now regard is suitable for publication on Cancers.

Sincerely,

Francesco Moccia

Francesco Moccia, PhD

Laboratory of General Physiology,

Department of Biology and Biotechnology “L. Spallanzani”

University of Pavia,

Via Forlanini 6, 27100, Pavia, Italy.

Tel: 0039 0382 987614.

Fax: 0039 0382 987527.

Reviewer 2 Report

Thanks for the updates. Please double-check to homogenous the format throughout the manuscript again before publication; the following lists are the examples.  Good luck.

1) Line 134 and line 179: Please consistent “1x106” or “1x10^6”.

2) Line 137: Please subscript “2” for “CO2”.

3) Line 200: Please use “45 minutes” to be consistent.

4) Line 211: Please be consistent with “minutes” or “min” throughout the manuscript.

5) Line 323, line 336, line 421: Please use italic p.

6) Line 239 and line 326: Please double-check and be consistent.

        7) Line 370: Please capitalized “ANOVA” throughout the manuscript.

Author Response

Dear Reviewer #2,

We are gratefully thankful for your comments on our manuscript entitled: “Store-operated Ca2+ entry is up-regulated in tumour infiltrating lymphocytes from metastatic colorectal cancer patients” for publication as Research Article in Cancers – Special Issue Advances in Cancer Immunotherapy.

We amended the manuscript according to your suggestions.

More specifically:
1)    Line 134 and line 179: Please consistent “1x106” or “1x10^6”.
“1x10^6” has been replaced with “1x106”.

2) Line 137: Please subscript “2” for “CO2”.
“2” in CO2 has been subscripted.

3) Line 200: Please use “45 minutes” to be consistent.
We used minutes.

4) Line 211: Please be consistent with “minutes” or “min” throughout the manuscript.
We used “min” throughout the manuscript.

5) Line 323, line 336, line 421: Please use italic p.
We used the italic p.

6) Line 239 and line 326: Please double-check and be consistent.
We have double-checked, thank you!

7) Line 370: Please capitalized “ANOVA” throughout the manuscript.
“ANOVA” has been capitalized throughout the manuscript.

Once again, we do thank you for the careful evaluation of the manuscript. We do hope that you will now regard our manuscript suitable for publication on Cancers.

Sincerely,
Francesco Moccia

Francesco Moccia, PhD
Laboratory of General Physiology,
Department of Biology and Biotechnology “L. Spallanzani”
University of Pavia, 
Via Forlanini 6, 27100, Pavia, Italy.
Tel: 0039 0382 987614.
Fax: 0039 0382 987527. 
E-mail: [email protected]